# ROC Analyses Based on Measuring Evidence Using the Relative Belief Ratio

**DOI:** 10.3390/e24121710

**Published:** 2022-11-23

**Authors:** Luai Al-Labadi, Michael Evans, Qiaoyu Liang

**Affiliations:** 1Department of Mathematical and Computational Sciences, University of Toronto Mississauga, Mississauga, ON L5L 1C6, Canada; 2Department of Statistical Sciences, University of Toronto, Toronto, ON M5S 3G3, Canada

**Keywords:** ROC, AUC, optimal cutoff, statistical evidence, relative belief, binormal, mixture Dirichlet process

## Abstract

ROC (Receiver Operating Characteristic) analyses are considered under a variety of assumptions concerning the distributions of a measurement *X* in two populations. These include the binormal model as well as nonparametric models where little is assumed about the form of distributions. The methodology is based on a characterization of statistical evidence which is dependent on the specification of prior distributions for the unknown population distributions as well as for the relevant prevalence *w* of the disease in a given population. In all cases, elicitation algorithms are provided to guide the selection of the priors. Inferences are derived for the AUC (Area Under the Curve), the cutoff *c* used for classification as well as the error characteristics used to assess the quality of the classification.

## 1. Introduction

An ROC (Receiver Operating Characteristic) analysis is used in medical science to determine whether or not a real-valued diagnostic variable *X* for a disease or condition is useful. If the diagnostic indicates that an individual has the condition, then this will typically mean that a more expensive or invasive medical procedure is undertaken. So it is important to assess the accuracy of the diagnostic variable X. These methods have a wider class of applications but our terminology will focus on the medical context.

An approach to such analyses is presented here that is based on a characterization of statistical evidence and which incorporates all available information as expressed via prior probability distributions. For example, while p-values are often used in such analyses, there are questions concerning the validity of these quantities as characterizations of statistical evidence. As will be discussed, there are many advantages to the framework adopted here.

A common approach to the assessment of the diagnostic variable *X* is to estimate its AUC (Area Under the Curve), namely, the probability that an individual sampled from the diseased population will have a higher value of diagnostic variable *X* than an individual independently sampled from the nondiseased population. A good diagnostic should give a value of the AUC near 1 while a value near 1/2 indicates a poor diagnostic test (if the AUC is near 0, then the classification is reversed). It is possible, however, that a diagnostic with AUC ≈1 may not be suitable (see Examples 1 and 6). In particular, a cutoff value *c* needs to be selected so that if X>c, then an individual is classified as requiring the more invasive procedure. Inferences about the error characteristics for the combination (X,c), such as the false positive rate, etc., are also required.

This paper is concerned with inferences about the AUC, the cutoff *c* and the error characteristics of the classification based on a valid measure of evidence. A key aspect of the analysis is the *relevant prevalence w*. The phrase “relevant prevalence” means that *X* will be applied to a certain population, such as those patients who exhibit certain symptoms, and *w* represents the proportion of this subpopulation who are diseased. The value of *w* may vary by geography, medical unit, time, etc. To make a valid assessment of *X* in an application, it is necessary that the information available concerning *w* be incorporated. This information is expressed here via an elicited prior probability distribution for *w*, which may be degenerate at a single value if *w* is assumed known, or be quite diffuse when little is known about *w*. In fact, all unknown population quantities are given elicited priors. There are many contexts where data are available relevant to the value of *w* and this leads to a full posterior analysis for *w* as well as for the other quantities of interest. Even when such data are not available, however, it is still possible to take the prior for *w* into account so the uncertainties concerning *w* always play a role in the analysis and this is a unique aspect of the approach taken here.

While there are some methods available for the choice of *c*, these often do not depend on the prevalence *w* which is a key factor in determining the true error characteristics of (X,c) in an application, see [1,2,3,4,5]. So it is preferable to take *w* into account when considering the value of a diagnostic in a particular context. One approach to choosing *c* is to minimize some error criterion that depends on *w* to obtain $c_{opt}.$ As will be demonstrated in the examples, however, sometimes $c_{opt}$ results in a classification that is useless. In such a situation a suboptimal choice of *c* is required but the error characteristics can still be based on what is known about *w* so that these are directly relevant to the application.

Others have pointed out deficiencies in the AUC statistic and proposed alternatives. For example, it can be argued that taking into account the costs associated with various misclassification errors is necessary and that using the AUC is implicitly making unrealistic assumptions concerning these costs, see [6]. While costs are relevant, costs are not incorporated here as these are often difficult to quantify. Our goal is to express clearly what the evidence is saying about how good (X,c) is via an assessment of its error characteristics. With the error characteristics in hand, a user can decide whether or not the costs of misclassifications are such that the diagnostic is usable. This may be a qualitative assessment although, if numerical costs are available, these could be subsequently incorporated. The principle here is that economic or social factors be considered separately from what the evidence in the data says, as it is a goal of statistics to clearly state the latter.

The framework for the analysis is Bayesian as proper priors are placed on the unknown distribution $F_{ND}$ (the distribution of *X* in the nondiseased population), on $F_{D}$ (the distribution of *X* in the diseased population) and the prevalence w. In all the problems considered, elicitation algorithms are presented for how to choose these priors. Moreover, all inferences are based on the relative belief characterization of statistical evidence where, for a given quantity, evidence in favor (against) is obtained when posterior beliefs are greater (less) than prior beliefs, see Section 2.2 for discussion and [7]. So evidence is determined by how the data change beliefs. Section 2 discusses the general framework, defines relevant quantities and provides an outline for how specific relative belief inferences are determined. Section 3 develops the inferences for the quantities of interest for three contexts (1) *X* is an ordered discrete variable with and without constraints on $\left(F_{ND},F_{D}\right)$ (2) *X* is a continuous variable and $\left(F_{ND},F_{D}\right)$ are normal distributions (the *binormal model*) (3) *X* is a continuous variable and no constraints are placed on $(F_{ND},F_{D}).$

There is previous work on using Bayesian methods in ROC analyses. For example, a Bayesian analysis for the binormal model when there are covariates present is developed in [8]. An estimate of the ROC using the Bayesian bootstrap is discussed in [9]. A Bayesian semiparametric analysis using a Dirichlet mixture process prior is developed in [10,11]. The sampling regime where the data can be used for inference about the relevant prevalence and where a gold standard classifier is not assumed to exist is presented in [12]. Considerable discussion concerning the case where the diagnostic test is binary, covering the cases where there is and is not a gold standard test, as well as the situation where the goal is to compare diagnostic tests and to make inference about the prevalence distribution can be found in [13] and also see [14]. Application of an ROC analysis to a comparison of linear and nonlinear approaches to a problem in medical physics is in [15]. Further discussion of nonlinear methodology can be found in [16,17].

The contributions of this paper, that have not been covered by previous published work in this area, are as follows:(i)The primary contribution is to base all the inferences associated with an ROC analysis on a clear and unambiguous characterization of statistical evidence via the principle of evidence and the relative belief ratio. While Bayes factors are also used to measure statistical evidence, there are serious limitations on their usage with continuous parameters as priors are restricted to be of a particular form. The approach via relative belief removes such restrictions on priors and provides a unified treatment of estimation and hypothesis assessment problems. In particular, this leads directly to estimates of all the quantities of interest, together with assessments of the accuracy of the estimates, and a characterization of the evidence, whether in favor of or against a hypothesis, together with a measure of the strength of the evidence. Moreover, no loss functions are required to develop these inferences. The merits of the relative belief approach over others are more fully discussed in Section 2.2.(ii)A prior on the relevant prevalence is always used to determine inferences even when the posterior distribution of this quantity is not available. As such the prevalence always plays a role in the inferences derived here.(iii)The error in the estimate of the cut-off is always quantified as well as the errors in the estimates of the characteristics evaluated at the chosen cut-off. It is these characteristics, such as the sensitivity and specificity, that ultimately determine the value of the diagnostic test.(iv)The hypothesis $H_{0}:$ AUC >1/2 is first assessed and if evidence is found in favor of this, the prior is then conditioned on this event being true for inferences about the remaining quantities. Note that this is equivalent to conditioning the posterior on the event AUC > 1/2 when inferences are determined by the posterior but with relative belief inferences both the conditioned prior and conditioned posterior are needed to determine the inferences.(v)Precise conditions are developed for the existence of an optimal cutoff with the binormal model.(vi)In the discrete context (1), it is shown how to develop a prior and the analysis under the assumption that the probabilities describing the outcomes from the diagnostic variable *X* are monotone.

The relative belief ratio, as a measure of evidence, is seen to have a connection to relative entropy. For example, it is equivalent, in the sense that the inferences are the same, to use the logarithm of the relative belief ratio as the measure of evidence. The relative entropy is then the posterior expectation of this quantity and so can be considered as a measure of the overall evidence provided by the model, prior and data concerning a quantity of interest.

The methods used for all the computations in the paper are simulation based and represent fairly standard Bayesian computational methods. In each context considered, sufficient detail is provided so that these can be implemented by a user.

## 2. The Problem

Consider the formulation of the problem as presented in [18,19] but with somewhat different notation. There is a measurement $X:\Omega \to R^{1}$ defined on a population $\Omega =\Omega_{D}\cup \Omega_{ND},$ with $\Omega_{D}\cap \Omega_{ND}=\phi ,$ where $\Omega_{D}$ is comprised of those with a particular disease, and $\Omega_{ND}$ represents those without the disease. So $F_{ND}\left(c\right)=\#\left(\left\{\omega \in \Omega_{ND}:X\left(\omega \right)\leq c\right\}\right)/\#\left(\Omega_{ND}\right)\;$ is the conditional cdf of *X* in the nondiseased population, and $F_{D}\left(x\right)=\#\left(\left\{\omega \in \Omega_{D}:X\left(\omega \right)\leq x\right\}\right)/\#\left(\Omega_{D}\right)$ is the conditional cdf of *X* in the diseased population. It is assumed that there is a gold standard classifier, typically much more difficult to use than X, such that for any ω∈Ω it can be determined definitively if $\omega \in \Omega_{D}$ or $\omega \in \Omega_{ND}.$ There are two ways in which one can sample from Ω, namely,

(i)take samples from each of $\Omega_{D}$ and $\Omega_{ND}$ separately or(ii)take a sample from Ω.

The sampling method used affects the inferences that can be drawn. For many studies (i) is the relevant sampling mode, as in case-control studies, while (ii) is relevant in cross-sectional studies.

It supposed that the greater the value X(ω) is for individual ω, the more likely it is that $\omega \in \Omega_{D}.$ For the classification, a cutoff value *c* is required such that, if X(ω)>c, then ω is classified as being in $\Omega_{D}$ and otherwise is classified as being in $\Omega_{ND}.$ However, *X* is an imperfect classifier for any *c* and it is necessary to assess the performance of (X,c). It seems natural that a value of *c* be used that is optimal in some sense related to the error characteristics of this classification. Table 1 gives the relevant probabilities for classification into $\Omega_{D}$ and $\Omega_{ND}$, together with some common terminology, in a *confusion matrix*.

Another key ingredient is the prevalence $w=\#\left(\Omega_{D}\right)/\#\left(\Omega \right)$ of the disease in Ω. In practical situations, it is necessary to also take *w* into account in assessing the error in (X,c). The following error characteristics depend on w,
$$\begin{array}{cc} \mathrm{Error}(c) & =\mathrm{misclassification}\;\mathrm{rate}=w\mathrm{FNR}(c)+(1-w)\mathrm{FPR}(c),\\ \mathrm{FDR}(c) & =\mathrm{false}\;\mathrm{discovery}\;\mathrm{rate}=\frac{\left(1-w)\mathrm{FPR}(c\right)}{w(1-\mathrm{FNR}(c))+(1-w)\mathrm{FPR}(c)},\\ \mathrm{FNDR}(c) & =\mathrm{false}\;\mathrm{nondiscovery}\;\mathrm{rate}=\frac{w\mathrm{FNR}(c)}{w\mathrm{FNR}(c)+(1-w)(1-\mathrm{FPR}(c))}. \end{array}$$Under sampling regime (ii) and cutoff *c*, Error(*c*) is the probability of making an error, FDR(*c*) is the conditional probability of a subject being misclassified as positive given that it has been classified as positive and FNDR(*c*) is the conditional probability of a subject being misclassified as negative given that it has been classified as negative. In other words, FDR(*c*) is the proportion of those individuals in the population consisting of those who have been classified by the diagnostic test as having the disease, but in fact do not have it. It is often observed that when *w* is very small and FNR(*c*) and FPR(*c*) are small, then FDR(*c*) can be big. This is sometimes referred to as the *base rate fallacy* as, even though the test appears to be a good one, there is a high probability that an individual classified as having the disease will be misclassified. For example, if w= FNR(*c*) = FPR(c)=0.05, then Error(c)=0.05, FDR(c)=0.50, FNDR$\left(c\right)=2.76\times 10^{-3}$ and when w=0.01, then Error(c)=0.05, FDR(c)=0.84, FNDR$\left(c\right)=5.31\times 10^{-4}.$ In these cases the false nondiscovery rate is quite small while the false discovery rate is large. If the disease is highly contagious, then these probabilities may be considered acceptable but indeed they need to be estimated. Similarly, FNDR(c) may be small when FNR(c) is large and *w* is very small.

It is naturally desirable to make inference about an optimal cutoff $c_{opt}$ and its associated error quantities. For a given value of w, the optimal cutoff will be defined here as $c_{opt}=\arg\inf$ Error(c), the value which minimizes the probability of making an error. Other choices for determining a $c_{opt}$ can be made, and the analysis and computations will be similar, but our thesis is that, when possible, any such criterion should involve the prior distribution of the relevant prevalence w. As demonstrated in Example 6 this can sometimes lead to useless values of $c_{opt}$ even when the AUC is large. While this situation calls into question the value of the diagnostic, a suboptimal choice of *c* can still be made according to some alternative methodology. For example, sometimes *Youden’s index*, which maximizes 1−2Error(c) over *c* with w=1/2, is recommended, or the *closest-to-(0,1)* criterion which minimizes $\mathrm{FPR}{\left(c\right)}^{2}+{\left(1-\mathrm{TPR}\left(c\right)\right)}^{2}$, see [2] for discussion. Youden’s index and the closest-to-(0,1) criterion do not depend on the prevalence and have geometrical interpretations in terms of the ROC curve, but as we will see, the ROC curve does not exist in full generality and this is particularly relevant in the discrete case. The methodology developed here provides an estimate of the *c* to be used, together with an exact assessment of the error in this estimate, as well as providing estimates of the associated error characteristics of the classification.

Letting ${\hat{c}}_{opt}$ denote the estimate of $c_{opt}$, the values of Error$\left({\hat{c}}_{opt}\right),\mathrm{TPR}\left({\hat{c}}_{opt}\right),\mathrm{FPR}\left({\hat{c}}_{opt}\right),\mathrm{FNR}\left({\hat{c}}_{opt}\right)$ and $\mathrm{TNR}({\hat{c}}_{opt})$ are also estimated and the recorded values used to assess the value of the diagnostic test. There are also other characteristics that may prove useful in this regard such as the *positive predictive value* (PPV)
$$\mathrm{PPV}\left(c\right)=\frac{w\mathrm{TPR}(c)}{w\mathrm{TPR}(c)+(1-w)\mathrm{FPR}(c)},$$
namely, the conditional probability a subject is positive given that they have tested positive, which plays a role similar to FDR(c). See [14] for discussion of the PPV and the similarly defined *negative predictive value* (NPV). The value of $\mathrm{PPV}({\hat{c}}_{opt})$ can be estimated in the same way as the other quantities as is subsequently discussed.

### 2.1. The AUC and ROC

Consider two situations where $F_{ND},F_{D}$ are either both absolutely continuous or both discrete. In the discrete case, suppose that these distributions are concentrated on a set of points $c_{1}<c_{2}<\ldots <c_{m}.$ When $\omega_{D},\omega_{ND}$ are selected using sampling scheme (i), then the probability that a higher score is received on diagnostic *X* by a diseased individual than a nondiseased individual is
(1)$$\mathrm{AUC}=\left\{\begin{array}{cc} \int_{-\infty }^{\infty }\left(1-F_{D}\left(c\right)\right)\;f_{ND}\left(c\right)\;dc & \mathrm{abs}.\mathrm{cont}.\\ \sum_{i=1}^{m}\left(1-F_{D}\left(c_{i}\right)\right)\left(F_{ND}\left(c_{i}\right)-F_{ND}\left(c_{i-1}\right)\right) & \mathrm{discrete}. \end{array}\right.$$Under the assumption that $F_{D}\left(c\right)$ is constant on $\left\{c:F_{ND}\left(c\right)=p\right\}$ for every p∈[0,1], there is a function ROC (*receiver operating characteristic*) such that $1-F_{D}\left(c\right)=$ ROC$\left(1-F_{ND}\left(c\right)\right)$ so AUC$=\int_{-\infty }^{\infty }$ROC$\left(1-F_{ND}\left(c\right)\right)\;F_{ND}\left(dx\right).$ Putting $p=1-F_{ND}\left(c\right),$ then ROC$\left(p\right)=1-F_{D}\left(F_{ND}^{-1}\left(1-p\right)\right).\;$ In the absolutely continuous case, AUC$=\int_{0}^{1}$ROC(p)dp which is the *area under the curve* given by the ROC function. The area under the curve interpretation is geometrically evocative but is not necessary for (Equation 1) to be meaningful.

It is commonly suggested that a good diagnostic variable *X* will have an AUC close to 1 while a value close to 1/2 suggests a poor diagnostic test. It is surely the case, however, that the utility of *X* in practice will depend on the cutoff *c* chosen and the various error characteristics associated with this choice. So while the AUC can be used to screen diagnostics, it is only part of the analysis and inferences about the error characteristics are required to truly assess the performance of a diagnostic. Consider an example.

**Example 1.** 
*Suppose that $F_{D}=F_{ND}^{q}$ for some q>1, where $F_{ND}$ is continuous, strictly increasing with associated density $f_{ND}.$ Then using (Equation 1), AUC =1−1/(q+1) which is approximately 1 when q is large. The optimal c minimizes Error$\left(c\right)=wF_{ND}^{q}\left(c\right)+\left(1-w\right)\left(1-F_{ND}\left(c\right)\right)$ which implies c satisfies $F_{ND}\left(c\right)={\left\{\left(1-w\right)/qw\right\}}^{1/(q-1)}$ when q>(1−w)/w and the optimal c is otherwise c=∞. If q=99, then AUC =0.99 and with w=0.025,(1−w)/w=39<q so FNR$(c_{opt})=0.390,$ FPR$(c_{opt})=0.009,$ Error$(c_{opt})=0.019,$ FDR$(c_{opt})=0.009$ and FNDR$(c_{opt})=0.010$. So X seems like a good diagnostic via the AUC and the error characteristics that depend on the prevalence although within the diseased population the probability is 0.39 of not detecting the disease. If instead w=0.01, then the AUC is the same but q=99=(1−w)/w and the optimal classification always classifies an individual as non-diseased which is useless. So the AUC does not indicate enough about the characteristics of the diagnostic to determine if it is useful or not. It is necessary to look at the error characteristics of the classification at the cutoff value that will actually be used, to determine if a diagnostic is suitable and this implies that information about w is necessary in an application.*


### 2.2. Relative Belief Inferences

Suppose there is a model $\left\{f_{\theta }:\theta \in \Theta \right\}$ for data x together with a prior probability measure Π, with density π, on Θ. These ingredients lead, via the *principle of conditional probability*, to beliefs about the true value of θ, as initially expressed by Π, being replaced by the posterior probability measure Π(·|x) with density π(·|x). Note that if interest is instead in a quantity ψ=Ψ(θ), where Ψ:Θ→Ψ and we use the same notation for the function and its range, then the model is replaced by $\{m_{\psi }:\psi \in \Psi \},$ where $m_{\psi }\left(x\right)=\int_{\Psi^{-1}\left\{\psi \right\}}f_{\theta }\left(x\right)\pi \left(\theta \;|\;\psi \right)\;d\theta$ is obtained by integrating out the nuisance parameters, and the prior is replaced by the marginal prior $\pi_{\Psi }\left(\psi \right)=\int_{\Psi^{-1}\left\{\psi \right\}}\pi \left(\theta \right)\;d\theta .$ This leads to the marginal posterior $\Pi_{\Psi }\left(\cdot \;|\;x\right)$ with density $\pi_{\Psi }\left(\cdot \;|\;x\right).$

For the moment suppose that all the distributions are discrete. The *principle of evidence* then says that there is evidence in favor of the value ψ if $\pi_{\Psi }\left(\psi \;|\;x\right)>\pi_{\Psi }\left(\psi \right),$ evidence against the value ψ if $\pi_{\Psi }\left(\psi \;|\;x\right)<\pi_{\Psi }\left(\psi \right),$ and no evidence either way if $\pi_{\Psi }\left(\psi \;|\;x\right)=\pi_{\Psi }\left(\psi \right).$ So, for example, there is evidence in favor of ψ if the probability of ψ increases after seeing the data. To order the possible values with respect to the evidence, we use the *relative belief ratio*
$$RB_{\Psi }\left(\psi \;|\;x\right)=\frac{\pi_{\Psi }\left(\psi \;|\;x\right)}{\pi_{\Psi }\left(\psi \right)}.$$Note that $RB_{\Psi }\left(\psi \;|\;x\right)>\left(<\right)1$ indicates whether there is evidence in favor of (against) the value ψ. If there is evidence in favor of both $\psi_{1}$ and $\psi_{2},$ then there is more evidence in favor of $\psi_{1}$ than $\psi_{2}$ whenever $RB_{\Psi }\left(\psi_{1}\;|\;x\right)>RB_{\Psi }\left(\psi_{2}\;|\;x\right)$ and, if there is evidence against both $\psi_{1}$ and $\psi_{2},$ then there is more evidence against $\psi_{1}$ than $\psi_{2}$ whenever $RB_{\Psi }\left(\psi_{1}\;|\;x\right)<RB_{\Psi }\left(\psi_{2}\;|\;x\right).$ For the continuous case consider a sequence of neighborhoods $N_{\epsilon }\left(\psi \right)↓\left\{\psi \right\}$ as ϵ→0 and then
(2)$$RB_{\Psi }\left(N_{\epsilon }\left(\psi \right)\;|\;x\right)=\frac{\Pi_{\Psi }\left(N_{\epsilon }\left(\psi \right)\;|\;x\right)}{\Pi_{\Psi }\left(N_{\epsilon }\left(\psi \right)\right)}\to \frac{\pi_{\Psi }\left(\psi \;|\;x\right)}{\pi_{\Psi }\left(\psi \right)}$$
under very weak conditions such as $\pi_{\Psi }\left(\psi \right)>0$ and $\pi_{\Psi }$ being continuous at ψ.

All the inferences about quantities considered in the paper are derived based upon the principle of evidence as expressed via the relative belief ratio. For example, it is immediate that the value $RB_{\Psi }\left(\psi_{0}\;|\;x\right)$ indicates whether or not there is evidence in favor of or against the hypothesis $H_{0}:\Psi \left(\theta \right)=\psi_{0}.$ Furthermore, the posterior probability $\Pi_{\Psi }\left(RB_{\Psi }\left(\psi \;|\;x\right)\leq RB_{\Psi }\left(\psi_{0}\;|\;x\right)\;|\;x\right)$ measures the strength of this evidence for, if $RB_{\Psi }\left(\psi_{0}\;|\;x\right)>1$ and this probability is large, then there is strong evidence in favor of $H_{0}$ as there is a small belief that the true value has a larger relative belief ratio and if $RB_{\Psi }\left(\psi_{0}\;|\;x\right)<1$ and this probability is small, then there is strong evidence against $H_{0}$ as there is high belief that the true value has a larger relative belief ratio. For estimation it is natural to estimate ψ by the *relative belief estimate*
$\psi \left(x\right)=\arg\sup_{\psi \in \Psi }RB_{\Psi }\left(\psi \;|\;x\right)$ as this value has the maximum evidence in its favor. Furthermore, the accuracy of this estimate can be assessed by looking at the *plausible region* $Pl_{\Psi }\left(x\right)=\left\{\psi :RB_{\Psi }\left(\psi \;|\;x\right)>1\right\},$ consisting of all those values for which there is evidence in favor, together with its size and posterior content which measures how strongly it is believed the true value lies in this set. Rather than using the plausible region to assess the accuracy of ψ(x), one could quote a *γ−relative belief credible region*
$$C_{\Psi ,\gamma }\left(x\right)=\left\{\psi :RB_{\Psi }\left(\psi \;|\;x\right)>c_{\gamma }\right\}$$
where the constant $c_{\gamma }$ is the largest value such that $\Pi_{\Psi }\left(C_{\Psi ,\gamma }\left(x\right)\;|\;x\right)\geq \gamma$. It is necessary, however, that $\gamma \leq \Pi_{\Psi }\left(Pl_{\Psi }\left(x\right)\;|\;x\right)$ as otherwise $C_{\Psi ,\gamma }\left(x\right)$ will contain values for which there is evidence against, and this is only known after the data have been seen.

It is established in [7], and in papers referenced there, that these inferences possess a number of good properties such as consistency, satisfy various optimality criteria and clearly they are based on a direct measure of the evidence. Perhaps most significant is the fact that all the inferences are invariant under reparameterizations. For if λ=Λ(ψ), where Λ is a smooth bijection, then
$$RB_{\Lambda }\left(\lambda \;|\;x\right)=\frac{\pi_{\Lambda }\left(\lambda \;|\;x\right)}{\pi_{\Lambda }\left(\lambda \right)}=\frac{\pi_{\Psi }\left(\Lambda^{-1}\left(\lambda \right)\;|\;x\right)J_{\Lambda }\left(\Lambda^{-1}\left(\lambda \right)\right)}{\pi_{\Psi }(\Lambda^{-1}\left(\lambda \right))J_{\Lambda }(\Lambda^{-1}\left(\lambda \right))}=\frac{\pi_{\Psi }(\left(\psi \right)\;\left|\;x\right)}{\pi_{\Psi }\left(\psi \right)}$$
and so, for example, λ(x)=Λ(ψ(x)). This invariance property is not possessed by the most common inference methods employed such as MAP estimation or using posterior means and this invariance holds no matter what the dimension of ψ is. Moreover, it is proved in [20] that relative belief inferences are optimally robust among all Bayesian inferences for ψ, to linear contaminations of the prior on ψ.

An analysis, using relative belief, of the data obtained in several physics experiments that were all concerned with examining whether there was evidence in favor of or against the quantum model versus hidden variables is available in [21]. Furthermore, an approach to checking models used for quantum mechanics via relative belief is discussed in [22]. Other applications of relative belief inferences to common problems of statistical practice can be found in [7].

The Bayes factor is an alternative measure of evidence and is commonly used for hypothesis assessment in Bayesian inference. To see why the relative belief ratio has advantages over the Bayes factor for evidence-based inferences consider first assessing the hypothesis $H_{0}:\Psi \left(\theta \right)=\psi_{0}.$ When the prior probability of $\psi_{0}$ satisfies $0<\Pi_{\Psi }\left(\left\{\psi_{0}\right\}\right)<1,$ then the Bayes factor is defined as the ratio of the posterior odds in favor of $H_{0}$ to the prior odds in favor of $H_{0},$ namely,
$$BF_{\Psi }\left(\psi_{0}\;|\;x\right)=\left\{\frac{\Pi_{\Psi }\left(\left\{\psi_{0}\right\}\;|\;x\right)}{\Pi_{\Psi }\left({\left\{\psi_{0}\right\}}^{c}\;|\;x\right)}\right\}{\left\{\frac{\Pi_{\Psi }\left(\left\{\psi_{0}\right\}\right)}{\Pi_{\Psi }\left({\left\{\psi_{0}\right\}}^{c}\right)}\right\}}^{-1}.$$It is easily shown that the Bayes factor satisfies the principle of evidence and $BF_{\Psi }\left(\psi_{0}\;|\;x\right)>\left(<\right)1$ is evidence in favor (against) $H_{0},$ so in this context it is a valid measure of evidence.

One might wonder why it is necessary to consider a ratio of odds as opposed to the simpler ratio of probabilities, as specified by the relative belief ratio, for the purpose of measuring evidence but in fact there is a more serious issue with the Bayes factor. For suppose, as commonly arises in applications, that $\Pi_{\Psi }$ is a continuous probability measure so that $\Pi_{\Psi }(\left\{\psi_{0}\right\})=0$ as then the Bayes factor for $H_{0}$ is not defined. The common recommendation in this context is to require the specification of the following ingredients: a prior probability p>0, a prior distribution $\Pi_{H_{0}}$ concentrated on $\Psi^{-1}\left\{\psi_{0}\right\}$ which provides the prior predictive density $m_{H_{0}}\left(x\right)$, a prior distribution $\Pi_{H_{0}^{c}}$ concentrated on $\Psi^{-1}{\left\{\psi_{0}\right\}}^{c}$ which provides the prior predictive density $m_{H_{0}^{c}}\left(x\right)$ and then the full prior is taken to be the mixture $\Pi =p\Pi_{H_{0}}+\left(1-p\right)\Pi_{H_{0}^{c}}.$ With this prior the Bayes factor for $H_{0}$ is defined, as now the prior probability of $\psi_{0}$ equals p, and an easy calculation shows that $BF_{\Psi }\left(\psi_{0}\;|\;x\right)=m_{H_{0}}\left(x\right)/m_{H_{0}^{c}}\left(x\right).$ Typically the prior $\Pi_{H_{0}^{c}}$ is taken to be the prior that we might place on θ when interest is in estimating ψ.

Now consider the problem of estimating ψ and the prior is such that $\Pi_{\Psi }(\left\{\psi \right\})=0$ for every value of ψ as with a continuous prior. The Bayes factor is then not defined for any value of ψ and, if we wished to use the Bayes factor for estimation purposes, it would be necessary to modify the prior to be a different mixture for each value of ψ so that there would be in effect multiple different priors. This does not correspond to the logic underlying Bayesian inference. When using the relative belief ratio for inference only one prior is required and the same measure of evidence is used for both hypothesis assessment and estimation purposes.

Another approach to dealing with the problem that arises with the Bayes factor and continuous priors is to take a limit as in (Equation 2) and, when this is done, we obtain the result
$$BF_{\Psi }\left(N_{\epsilon }\left(\psi \right)\;|\;x\right)\to RB_{\Psi }\left(\psi \;|\;x\right)$$
as ϵ→0 whenever the prior density of Ψ is continuous and positive at ψ. In other words the relative belief ratio can be also considered as a natural definition of the Bayes factor in continuous contexts.

## 3. Inferences for an ROC Analysis

Suppose we have a sample of $n_{D}$ from $\Omega_{D}$, namely, $x_{D}=\left(x_{D1},\ldots ,x_{Dn_{D}}\right)$ and a sample of $n_{ND}$ from $\Omega_{ND}$, namely, $x_{ND}=\left(x_{ND1},\ldots ,x_{NDn_{ND}}\right)$ and the goal is to make inference about the AUC, the cutoff *c* and the error characteristics FNR(c), FPR(c), Error(c), FDR(c) and FNDR(c). For the AUC it makes sense to first assess the hypothesis $H_{0}:$ AUC >1/2 via stating whether there is evidence for or against $H_{0}$ together with an assessment of the strength of this evidence. Estimates are required for all of these quantities, together with an assessment of the accuracy of the estimate.

### 3.1. The Prevalence

Consider first inferences for the relevant prevalence w. If *w* is known, or at least assumed known, then nothing further needs to be done but otherwise this quantity needs to be estimated when assessing the value of the diagnostic and so uncertainty about *w* needs to be addressed.

If the full data set is based on sampling scheme (ii), then $n_{D}∼$ binomial(n,w). A natural prior $\pi_{W}$ to place on *w* is a beta$\left(\alpha_{1w},\alpha_{2w}\right)$ distribution. The hyperparameters are chosen based on the elicitation algorithm discussed in [23] where interval [l,u] is chosen such that it is believed that w∈[l,u] with prior probability γ. Here [l,u] is chosen so that we are virtually certain that w∈[l,u] and γ=0.99 then seems like a reasonable choice. Note that choosing l=u corresponds to *w* being known and so γ=1 in that case. Next pick a point $\xi_{w}\in \left[l,u\right]$ for the mode of the prior and a reasonable choice might be $\xi_{w}=\left(l+u\right)/2.$ Then putting $\tau_{w}=\alpha_{1w}+\alpha_{2w}-2$ leads to the parameterization beta$(\alpha_{1w},\alpha_{2w})=$ beta$\left(1+\tau_{w}\xi_{w},1+\tau_{w}\left(1-\xi_{w}\right)\right)$ where $\xi_{w}$ locates the mode and $\tau_{w}$ controls the spread of the distribution about $\xi_{w}.$ Here $\tau_{w}=0$ gives the uniform distribution and $\tau_{w}=\infty$ gives the distribution degenerate at $\xi_{w}.$ With $\xi_{w}$ specified, $\tau_{w}$ is the smallest value of $\tau_{w}$ such that the probability content of [l,u] is γ and this is found iteratively. For example, if [l,u]=[0.60,0.70] and γ=0.99, so *w* is known reasonably well, then $\xi_{w}=\left(l+u\right)/2=0.65$ and $\tau_{w}=601.1,$ so the prior is beta(391.72,211.39) and the posterior is beta$(391.72+n_{D},211.39+n_{ND}).$

The estimate of *w* is then
$$w\left(n_{D},n_{ND}\right)=\arg\underset{w\in [0,1]}{\sup}RB\left(w\;|\;n_{D},n_{ND}\right)=\arg\underset{w\in [0,1]}{\sup}\frac{\pi_{W}\left(w\;|\;n_{D},n_{ND}\right)}{\pi_{W}\left(w\right)}.$$In this case the estimate is the MLE, namely, $w\left(n_{D},n_{ND}\right)=n_{D}/\left(n_{D}+n_{ND}\right).$ The accuracy of this estimate is measured by the size of the plausible region $Pl\left(n_{D},n_{ND}\right)=\left\{w:RB\left(w\;|\;n_{D},n_{ND}\right)>1\right\}$. For example, if n=100 and $n_{D}=68,$ then w(68,32)=0.68 and Pl(68,32)=[0.647,0.712] which has posterior content 0.651. So the data suggest that the upper bound of u=0.70 is too strong although the posterior belief in this interval is not very high.

The prior and posterior distributions of *w* play a role in inferences about all the quantities that depend on the prevalence. In the case where the cutoff is determined by minimizing the probability of a misclassification, then $c_{opt},$ FNR$(c_{opt}),$ FPR$(c_{opt}),$ Error$(c_{opt}),$ FDR$\left(c_{opt}\right)$ and FNDR$\left(c_{opt}\right)$ all depend on the prevalence. Under sampling scheme (i), however, only the prior on *w* has any influence when considering the effectiveness of X. Inference for these quantities is now discussed in both cases.

### 3.2. Ordered Discrete Diagnostic

Suppose *X* takes values on the finite ordered scale $c_{1}<c_{2}<\ldots <c_{m}$ and let $p_{NDi}=P\left(X\left(\omega_{ND}\right)=c_{i}\right),p_{Di}=P\left(X\left(\omega_{D}\right)=c_{i}\right)$ so $F_{ND}\left(c_{i}\right)=\sum_{j=1}^{i}p_{NDj}$ and $F_{D}\left(c_{i}\right)=\sum_{j=1}^{i}p_{Dj}$. These imply that FPR$\left(c_{i}\right)=1-\sum_{j=1}^{i}p_{NDi},$ FNR$\left(c_{i}\right)=\sum_{j=1}^{i}p_{Di},$
$$\mathrm{AUC}\left(p_{ND},p_{D}\right)=\sum_{i=1}^{m}\left(1-\mathrm{FNR}(c_{i})\right)p_{NDi}$$
with the remaining quantities defined similarly. Ref. [23] can be used to obtain independent elicited Dirichlet priors
(3)$$p_{ND}∼\mathrm{Dirichlet}\left(\alpha_{ND1},\ldots ,\alpha_{NDm}\right),\;p_{D}∼\mathrm{Dirichlet}\left(\alpha_{D1},\ldots ,\alpha_{Dm}\right)$$
on these probabilities by placing either upper or lower bounds on each cell probability that hold with virtual certainty γ, as discussed for the beta prior on the prevalence. If little information is available, it is reasonable to use uniform (Dirichlet(1,…,1)) priors on $p_{ND}$ and $p_{D}.$ This together with the independent prior on *w* leads to prior distributions for the AUC, $c_{opt}$ and all the quantities associated with error assessment such as FNR$(c_{opt}),$ etc.

Data $\left(x_{D},x_{ND}\right)$ lead to counts $f_{ND}=\left(f_{ND1},\ldots ,f_{NDm}\right)$ and $f_{D}=\left(f_{D1},\ldots ,f_{Dm}\right)$ which in turn lead to the independent posteriors
(4)$$p_{ND}\;\left|\;f_{ND}∼\mathrm{Dirichlet}\left(\alpha_{ND}+f_{ND}\right),\;p_{D}\;\right|\;f_{D}∼\mathrm{Dirichlet}\left(\alpha_{D}+f_{D}\right).$$Under sampling regime (ii) this, together with the independent posterior on w, leads to posterior distributions for all the quantities of interest. Under sampling regime (i), however, the logical thing to do, so the inferences reflect the uncertainty about w, is to only use the prior on *w* when deriving inferences about any quantities that depend on this such as $c_{opt}$ and the various error assessments.

Consider inferences for the AUC. The first inference should be to assess the hypothesis $H_{0}:$ AUC >1/2 for, if $H_{0}$ is false, then *X* would seem to have no value as a diagnostic (the possibility that the directionality is wrong is ignored here). The relative belief ratio $RB\left(H_{0}\;|\;f_{ND},f_{ND}\right)=\Pi \left(H_{0}\;|\;f_{ND},f_{ND}\right)/\Pi \left(H_{0}\right)$ is computed and compared to 1. If it is concluded that $H_{0}$ is true, then perhaps the next inference of interest is to estimate the AUC via the relative belief estimate. The prior and posterior densities of the AUC are not available in closed form so estimates are required and density histograms are employed here for this. The set (0,1] is discretized into *L* subintervals $\left(0,1\right]=\cup_{i=1}^{L}(\left(i-1)/L,i/L\right],$ and putting $a_{i}=\left(i-1/2\right)/L,$ the value of the prior density $p_{\mathrm{AUC}}\left(a_{i}\right)$ is estimated by L( proportion of prior simulated values of AUC in (i−1,i]/L) and similarly for the posterior density $p_{\mathrm{AUC}}\left(a_{i}\;|\;f_{ND},f_{D}\right).$ Then $RB_{\mathrm{AUC}}\left(a\;|\;f_{ND},f_{ND}\right)$ is maximized to obtain the relative belief estimate AUC$\left(f_{ND},f_{D}\right)$ together with the plausible region and its posterior content.

These quantities are also obtained for $c_{opt}$ in a similar fashion, although $c_{opt}$ has prior and posterior distribution concentrated on $\left\{c_{1},c_{2},\ldots ,c_{m}\right\}$ so there is no need to discretize. Estimates of the quantities FNR$(c_{opt}\left(f_{ND},f_{D}\right)),$ FPR$(c_{opt}\left(f_{ND},f_{D}\right)),$ Error$(c_{opt}\left(f_{ND},f_{D}\right)),$ FDR$\left(c_{opt}\left(f_{ND},f_{D}\right)\right)$ and FNDR$\left(c_{opt}\left(f_{ND},f_{D}\right)\right)$ are also obtained as these indicate the performance of the diagnostic in practice. The relative belief estimates of these quantities are easily obtained in a second simulation where $c_{opt}\left(f_{ND},f_{D}\right)$ is fixed.

Consider now an example.

**Example 2.** 
*Simulated example.*

*For k=5 and $c_{i}=i$, data were generated as*

$$\begin{array}{cc} f_{ND} & ∼\mathrm{multinomial}\left(50,0.5,0.2,0.1,0.1,0.1\right)\;\mathrm{obtaining}\;f_{ND}=\left(29,7,4,5,5\right),\\ f_{D} & ∼\mathrm{multinomial}\left(100,0.1,0.1,0.2,0.3,0.3\right)\;\mathrm{obtaining}\;f_{D}=\left(14,7,25,33,21\right). \end{array}$$

*With these choices for $p_{ND},p_{D}$ the true values are AUC=0.65, and with w=0.65, $c_{opt}=2,$ FNR$(c_{opt})=0.200,$ FPR$(c_{opt})=0.300,$ Error${}_{w}\left(c_{opt}\right)=0.235,$ FDR$(c_{opt})=0.168$ and FNDR$(c_{opt})=0.347$. So X is not an outstanding diagnostic but with these error characteristics it may prove suitable for a given application. Uniform, namely, Dirichlet(1,1,1,1,1), priors were placed on $p_{ND}\;$ and $p_{D},$ reflecting little knowledge about these quantities.*

*Simulations based on Monte Carlo sample sizes of $N=10^{5}$ from the prior and posterior distributions of $p_{ND}\;$ and $p_{D}$ were conducted and the prior and posterior distributions of the quantities of interest obtained. The hypothesis $H_{0}:$ AUC >0.5 is assessed by $RB_{\mathrm{AUC}}\left(\left(0.50,1.00\right]\;|\;f_{ND},f_{D}\right)=3.15.$ So there is evidence in favor of $H_{0}$ and the strength of this evidence is measured by the posterior probability content of (0.50,1.00] which equals 1.0 to machine accuracy and so this is categorical evidence in favor of $H_{0}.$*

*For the continuous quantities a grid based on L+1=25 equispaced points {0,0.04,0.08,…,1.00} was used and all the mass in the interval (i−1,i]/L assigned to the midpoint (i−1/2)/L. Figure 1 contains plots of the prior and posterior densities and relative belief ratio of the AUC. The relative belief estimate of the AUC is AUC$(f_{ND},f_{D})=0.66$ with $Pl_{\mathrm{AUC}}\left(f_{ND},f_{D}\right)=\left[0.60,0.72\right]$ having posterior content 0.97. Certainly a finer partition of [0,1] than just 24 intervals is possible, but even in this relatively coarse case the results are quite accurate.*

*Supposing that the relevant prevalence is known to be w=0.65, Figure 2 contains plots of the prior and posterior densities and relative belief ratio of $c_{opt}.$ The relative belief estimate is $c_{opt}\left(f_{ND},f_{D}\right)=2$ with $Pl_{c_{opt}}\left(f_{ND},f_{D}\right)=\left\{2\right\}$ with posterior probability content 0.53 so the correct optimal cut-off has been identified but there is a degree of uncertainty concerning this. The error characteristics that tell us about the utility of X as a diagnostic are given by the relative belief estimates (column (a)) in Table 2. It is interesting to note that the estimate of Error$\left(c_{opt}\right)$ is determined by the prior and posterior distributions of a convex combination of FPR$\left(c_{opt}\right)$ and FNR$\left(c_{opt}\right)$ and the estimate is not the same convex combination of the estimates of FPR$\left(c_{opt}\right)$ and FNR$\left(c_{opt}\right)$. So, in this case Error$\left(c_{opt}\right)$ seems like a much better assessment of the performance of the diagnostic.*

*Suppose now that the prevalence is not known but there is a beta$\left(1+\tau_{w}\xi_{w},1+\tau_{w}\left(1-\xi_{w}\right)\right)$ prior specified for w and consider the choice discussed in Section 3.1 where $\xi_{w}=0.65$ and $\tau_{w}=601.1.$ When the data are produced according to sampling regime (i), then there is no posterior for w but this prior can still be used in determining the prior and posterior distributions of $c_{opt}$ and the associated error characteristics. When this simulation was carried out $c_{opt}\left(f_{ND},f_{D}\right)=2$ with $Pl_{c_{opt}}\left(f_{ND},f_{D}\right)=\left\{2\right\}$ with posterior probability content 0.53. and column (b) of Table 2 gives the estimates of the error characteristics. So other than the estimate of the FPR, the results are similar. Finally, assuming that the data arose under sampling scheme (ii), then w has a posterior distribution and using this gives $c_{opt}\left(f_{ND},f_{D}\right)=2$ with $Pl_{c_{opt}}\left(f_{ND},f_{D}\right)=\left\{2\right\}$ with posterior probability content 0.52 and error characteristics as in column (c) of Table 2. These results are the same as if the prevalence is known which is sensible as the posterior concentrates about the true value more than the prior.*

*Another somewhat anomalous feature of this example is the fact that uniform priors on $p_{D}$ and $p_{ND}$ do not lead to a prior on the AUC that is even close to uniform. In fact one could say that this prior has a built-in bias against a diagnostic with AUC >1/2 and indeed most choices of $p_{D}$ and $p_{ND}$ will not satisfy this. Another possibility is to require $p_{ND1}\geq \ldots \geq p_{NDm}$ and $p_{D1}\leq \ldots \leq p_{Dm},$ namely, require monotonicity of the probabilities. A result in [22] implies that $p_{ND}$ satisfies this iff $p_{ND}=A_{k}\omega_{ND}$ where $\omega_{ND}\in S_{k},$ the standard (k−1)-dimensional simplex, and $A_{k}\in R^{k\times k}$ with i-ith row equal to (0,…,0,1/i,1/(i+1),…,1/k) and $p_{D}$ satisfies this iff $p_{D}=B_{k}\omega_{D}$ where $\omega_{D}\in S_{k}$ and $B_{k}=I_{k}^{*}A_{k}$ where $I_{k}^{*}$$\in R^{k\times k}$ contains all 0’s except for 1’s on the crossdiagonal. If $\omega_{ND}$ and $\omega_{D}$ are independent and uniform on $S_{k},$ then $p_{D}$ and $p_{ND}$ are independent and uniform on the sets of probabilities satisfying the corresponding monotonicities and Figure 3 has a plot of the prior of the AUC when this is the case. It is seen that this prior is biased in favor of AUC >1/2. Figure 3 also has a plot of the prior of the AUC when $p_{D}$ is uniform on the set of all nondecreasing probabilities and $p_{ND}$ is uniform on $S_{k}.$ This reflects a much more modest belief that X will satisfy AUC >1/2 and indeed this may be a more appropriate prior than using uniform distributions on $S_{k}.$ Ref. [22] also provides elicitation algorithms for choosing alternative Dirichlet distributions for $\omega_{ND}$ and $\omega_{D}.$*

*When $H_{0}:$ AUC >0.5 is accepted, it makes sense to use the conditional prior, given that this event is true, in the inferences. As such it is necessary to condition the prior on the event $\sum_{i=1}^{m}\left(\sum_{j=1}^{i}p_{Dj}\right)p_{NDi}\leq 1/2.$ In general, it is not clear how to generate from this conditional prior but depending on the size of m and the prior, a brute force approach is to simply generate from the unconditional prior and select those samples for which the condition is satisfied and the same approach works with the posterior.*

*Here m=5, and using uniform priors for $p_{ND}$ and $p_{D}$, the prior probability of AUC >0.5 is 0.281 while the posterior probability is 0.998 so the posterior sampling is much more efficient. Choosing priors that are more favorable to AUC >0.5 will improve the efficiency of the prior sampling. Using the conditional priors led to AUC$(f_{ND},f_{D})=0.66$ with $Pl_{\mathrm{AUC}}\left(f_{ND},f_{D}\right)=\left[0.60,0.76\right]$ with posterior content 0.85. This is similar to the results obtained using the unconditional prior but the conditional prior puts more mass on larger values of the AUC hence the wider plausible region with lower posterior content. Moreover, $c_{opt}\left(f_{ND},f_{D}\right)=2$ with $Pl_{c_{opt}}\left(f_{ND},f_{D}\right)=\left\{1,2\right\}$ with posterior probability content approximately 1.00 (actually 0.99999) which reflects virtual certainty that the true optimal value is in {1,2}.*


### 3.3. Binormal Diagnostic

Suppose now that *X* is a continuous diagnostic variable and it is assumed that the distributions $F_{D}$ and $F_{ND}$ are normal distributions. The assumption of normality should be checked by an appropriate test and it will be assumed here that this has been carried out and normality was not rejected. While the normality assumption may seem somewhat unrealistic, many aspects of the analysis can be expressed in closed form and this allows for a deeper understanding of ROC analyses more generally.

With Φ denoting the N(0,1) cdf, then FNR$\left(c\right)=\Phi \left(\left(c-\mu_{D}\right)/\sigma_{D}\right),\;$FPR$\left(c\right)=1-\Phi \left(\left(c-\mu_{ND}\right)/\sigma_{ND}\right)$ so $c=\mu_{ND}+\sigma_{ND}\Phi^{-1}\left(1-\left(1-F_{ND}\left(c\right)\right)\right)$ and
$$\mathrm{AUC}\;=\int_{-\infty }^{\infty }\Phi \left(\frac{\mu_{D}-\mu_{ND}}{\sigma_{D}}+\frac{\sigma_{ND}}{\sigma_{D}}z\right)\varphi \left(z\right)\;dz.$$For given $\left(\mu_{D},\sigma_{D},\mu_{ND},\sigma_{ND}\right)$ and c, all these values can be computed using Φ except the AUC and for that quadrature or simulation via generating z∼N(0,1) is required.

The following results hold for the AUC with the proofs in the Appendix A.
**Lemma 1.** *AUC >1/2 iff $\mu_{D}>\mu_{ND}$ and when $\mu_{D}>\mu_{ND},$ the AUC is a strictly increasing function of $\sigma_{ND}/\sigma_{D}.$*From Lemma 1 it is clear that it makes sense to restrict the parameterization so that $\mu_{D}>\mu_{ND}$ but we need to test the hypothesis $H_{0}:\mu_{D}>\mu_{ND}$ first. Clearly Error(c)=wFNR(c)+(1−w)FPR(c)→1−w as c→−∞ and Error(c)→w as c→∞ so, if Error(c) does not achieve a minimum at a finite value of c, then the optimal cut-off is infinite and the optimal error is min{w,1−w}. It is possible to give conditions under which a finite cutoff exists and express $c_{opt}$ in closed form when the parameters and the relevant prevalence *w* are all known.

**Lemma 2.** 
*(i) When $\sigma_{D}^{2}=\sigma_{ND}^{2}=\sigma^{2},$ then a finite optimal cut-off minimizing Error(c) exists iff $\mu_{D}>\mu_{ND}$ and in that case*

(5)
$$c_{opt}=\frac{\mu_{D}+\mu_{ND}}{2}+\frac{\sigma^{2}}{\mu_{D}-\mu_{ND}}\log\left(\frac{1-w}{w}\right).$$

*(ii) When $\sigma_{D}^{2}\neq \sigma_{ND}^{2},$ then a finite optimal cut-off exists iff*

(6)
$${\left(\mu_{D}-\mu_{ND}\right)}^{2}+2\left(\sigma_{D}^{2}-\sigma_{ND}^{2}\right)\log\left(\frac{1-w}{w}\frac{\sigma_{D}}{\sigma_{ND}}\right)\geq 0$$

*and in that case*


(7)$$c_{opt}=\frac{\sigma_{ND}^{2}\mu_{D}-\sigma_{D}^{2}\mu_{ND}}{\sigma_{ND}^{2}-\sigma_{D}^{2}}-\frac{\sigma_{ND}\sigma_{D}}{\sigma_{ND}^{2}-\sigma_{D}^{2}}{\left\{\begin{array}{c} {\left(\mu_{D}-\mu_{ND}\right)}^{2}+\\ 2\left(\sigma_{D}^{2}-\sigma_{ND}^{2}\right)\log\left(\frac{1-w}{w}\frac{\sigma_{D}}{\sigma_{ND}}\right) \end{array}\right\}}^{1/2}.$$Note that when w=1/2, then in (i) $c_{opt}=\left(\mu_{D}+\mu_{ND}\right)/2$ as one might expect. In the case of unequal variances there is an additional restriction beyond $\mu_{D}\geq \mu_{ND}$ required to hold if the diagnostic is to serve as a reasonable classifier. The following shows that these can be combined in a natural way.

**Corollary 1.** 
*The restrictions $\mu_{D}\geq \mu_{ND}$ and (Equation 6) hold iff*


(8)$$\mu_{D}-\mu_{ND}-{\left\{\max\left[0,-2\left(\sigma_{D}^{2}-\sigma_{ND}^{2}\right)\log\left(\frac{1-w}{w}\frac{\sigma_{D}}{\sigma_{ND}}\right)\right]\right\}}^{1/2}\geq 0.$$So, if one is unwilling to assume constant variance, then the hypothesis $H_{0}:$ (Equation 8) holds, needs to be assessed. There is some importance to these results as they demonstrate that a finite optimal cutoff may in fact not exist at least when considering both types of error. For example, when $\mu_{ND}=1,\mu_{D}=2,\sigma_{D}=1,\sigma_{ND}=1.5,$ then for any w≤0.30885, the optimal cutoff is $c_{opt}=\infty$ with Error(∞)=w. When $c_{opt}$ is infinite, then one may need to consider various cutoffs *c* and find one that is acceptable at least with respect to some of the error characteristics FNR(c), FPR(c), Error(c), FDR(c) and FNDR(c).

Consider now examples with equal and unequal variances.

**Example 3.** 
*Binormal with $\sigma_{ND}^{2}=\sigma_{D}^{2}.$*

*There may be reasons why the assumption of equal variance is believed to hold but this needs to be assessed and evidence in favor found. If evidence against the assumption is found, then the approach of Example 4 can be used. A possible prior is given by $\pi_{1}\left(\mu_{ND},\sigma^{2}\right)\pi_{2}\left(\mu_{D}\;|\;\sigma^{2}\right)$ where*

$$\mu_{ND}\;\left|\;\sigma^{2}∼N\left(\mu_{0},\tau_{0}^{2}\sigma^{2}\right),\;\mu_{D}\;\right|\;\sigma^{2}∼N\left(\mu_{0},\tau_{0}^{2}\sigma^{2}\right),\;1/\sigma^{2}∼\;\mathrm{gamma}\left(\lambda_{1},\lambda_{2}\right)$$

*which is a conjugate prior. The hyperparameters to be elicited are $(\mu_{0},\tau_{0}^{2},\lambda_{1},\lambda_{2}).$ Consider first eliciting the prior for $(\mu_{ND},\sigma^{2}).$ For this an interval $\left(m_{1},m_{2}\right)$ is specified such that is it believed that $\mu_{ND}\in \left(m_{1},m_{2}\right)$ with virtual certainty (say with probability γ=0.99). Then putting $\mu_{0}=\left(m_{1}+m_{2}\right)/2$ implies*

$$\gamma \leq \Phi \left(\left(m_{2}-\mu_{0}\right)/\tau_{0}\sigma \right)-\Phi \left(\left(m_{1}-\mu_{0}\right)/\tau_{0}\sigma \right)=2\Phi \left(\left(m_{2}-m_{1}\right)/2\tau_{0}\sigma \right)-1$$

*which implies $\sigma \leq \left(m_{2}-m_{1}\right)/2\tau_{0}z_{(1+\gamma )/2}$ where $z_{(1+\gamma )/2}=\Phi^{-1}\left(\left(1+\gamma \right)/2\right).$ The interval $\mu_{ND}\pm \sigma z_{(1+\gamma )/2}$ will contain an observation from $F_{ND\;}$ with virtual certainty and let $\left(l_{0},u_{0}\right)$ be lower and upper bounds on the half-length of this interval so $l_{0}/z_{(1+\gamma )/2}\leq \sigma \leq u_{0}/z_{(1+\gamma )/2}$ with virtual certainty. This implies $\tau_{0}=\left(m_{2}-m_{1}\right)/2u_{0}.$ This leaves specifying the hyperparameters $(\lambda_{1},\lambda_{2}),$ and letting $G(\cdot ,\lambda_{1},\lambda_{2})$ denote the cdf of the gamma$\left(\lambda_{1},\lambda_{2}\right)$ distribution, then $\left(\lambda_{1},\lambda_{2}\right)$ satisfying*

(9)
$$G\left(z_{(1+\gamma )/2}^{2}/l_{0}^{2},\lambda_{1},\lambda_{2}\right)=\left(1+\gamma \right)/2,\;G\left(z_{(1+\gamma )/2}^{2}/u_{0}^{2},\lambda_{1},\lambda_{2}\right)=\left(1-\gamma \right)/2$$

*will give the specified γ coverage. Noting that $G\left(x,\lambda_{1},\lambda_{2}\right)=G\left(\lambda_{2}x,\lambda_{1},1\right),$ first specify $\lambda_{1}$ and solve the first equation in (Equation 9) for $\lambda_{2}$ and then solve the second equation in (Equation 9) for $\lambda_{1}$ and continue this iteration until the probability content of $\left(l_{0}/z_{(1+\gamma )/2},u_{0}/z_{(1+\gamma )/2}\right)$ is sufficiently close to γ. Using $s_{D}^{2}=\left|\right|x_{D}-{\bar{x}}_{D}{1||}^{2},s_{ND}^{2}=\left|\right|x_{ND}-{\bar{x}}_{ND}1{\left|\right|}^{2},$ the posterior is then*

$$\begin{array}{cc} \; & \mu_{ND}\;|\;\sigma^{2},x_{ND}∼N\left({\left(n_{ND}+1/\tau_{0}^{2}\right)}^{-1}\left(n_{ND}{\bar{x}}_{ND}+\mu_{0}/\tau_{0}^{2}\right),{\left(n_{ND}+1/\tau_{0}^{2}\right)}^{-1}\sigma^{2}\right),\\ \; & \mu_{D}\;|\;\sigma^{2},x_{D}∼N\left({\left(n_{D}+1/\tau_{0}^{2}\right)}^{-1}\left(n_{D}{\bar{x}}_{D}+\mu_{0}/\tau_{0}^{2}\right),{\left(n_{D}+1/\tau_{0}^{2}\right)}^{-1}\sigma^{2}\right),\\ \; & 1/\sigma^{2}\;|\;\left(x_{ND},x_{D}\right)∼\;\mathrm{gamma}\left(\lambda_{1}+\left(n_{D}+n_{ND}\right)/2,\lambda_{x}\right) \end{array}$$

*where*

$$\begin{array}{ccc} \lambda_{x} & = & \lambda_{2}+\left(s_{D}^{2}+s_{ND}^{2}\right)/2+{\left(n_{D}+1/\tau_{0}^{2}\right)}^{-1}\left(n_{D}/\tau_{0}^{2}\right){\left({\bar{x}}_{D}-\mu_{0}\right)}^{2}/2+\\ & & {\left(n_{ND}+1/\tau_{0}^{2}\right)}^{-1}\left(n_{ND}/\tau_{0}^{2}\right){\left({\bar{x}}_{ND}-\mu_{0}\right)}^{2}/2. \end{array}$$

*Suppose the following values of the mss were obtained based on samples of $n_{ND}=25$ from $F_{ND}=N\left(0,1\right)$ and $n_{D}=20$ from $F_{D}=N\left(1,1\right)$*

$$\left({\bar{x}}_{ND},s_{ND}^{2}\right)=\left(-0.072,19.638\right),\;\left({\bar{x}}_{D},s_{D}^{2}\right)=\left(0.976,16.778\right).$$

*So the true values of the parameters are $\mu_{ND}=0,\mu_{D}=1,\sigma^{2}=1.$ In this case AUC $=\int_{-\infty }^{\infty }\Phi \left(1+z\right)\varphi \left(z\right)\;dz=0.760.$ Supposing that the relevant prevalence is $w=0.4,c_{opt}=0.5+\log\left(0.6/0.4\right)=0.905,$ FNR$\left(c_{opt}\right)=\Phi \left(0.905-1\right)=0.46$, FPR$\left(c_{opt}\right)=1-\Phi \left(0.905\right)=0.18$, Error$(c_{opt})=0.30$, FDR$(c_{opt})=0.34$, FNDR$(c_{opt})=0.27$,*

*For the prior elicitation, suppose it is known with virtual certainty that both means lie in (−5,5) and $\left(l_{0},u_{0}\right)=\left(1,10\right)$ so we take $\mu_{0}=\left(-5+5\right)/2=0,\tau_{0}=\left(m_{2}-m_{1}\right)/2u_{0}=0.5$ and the iterative process leads to $\left(\lambda_{1},\lambda_{2}\right)=\left(1.787,1.056\right).$ For inference about $c_{opt}$ it is necessary to specify a prior distribution for the prevalence w. This can range from w being completely known to being completely unknown whence a uniform(0,1) (beta(1,1)) would be appropriate. Following the developments of Section 3.1, suppose it is known that w∈[l,u]=[0.2,0.6] with prior probability γ=0.99, so in this case $\xi_{w}=\left(l+u\right)/2=0.4$ and $\tau_{w}=35.89725$ and the prior is w∼ beta(15.3589,22.53835).*

*The first inference step is to assess the hypothesis $H_{0}:$ AUC >1/2 which is equivalent to $H_{0}:$$\mu_{ND}<\mu_{D}$ by computing the prior and posterior probabilities of this event to obtain the relative belief ratio. The prior probability of $H_{0}$ given $\sigma^{2}$ is*

$$\int_{-\infty }^{\infty }\Phi \left(\left(\mu_{D}-\mu_{0}\right)/\tau_{0}\sigma \right){\left(\tau_{0}\sigma \right)}^{-1}\varphi \left(\left(\mu_{D}-\mu_{0}\right)/\tau_{0}\sigma \right)\;d\mu_{D}=1/2$$

*and averaging this quantity over the prior for $\sigma^{2}$ we get 1/2. The posterior probability of this event can be easily obtained via simulating from the joint posterior. When this is done in the specific numerical example, the relative belief ratio of this event is 2.011 with posterior content 0.999 so there is strong evidence that $H_{0}:$ AUC >1/2 is true.*

*If evidence is found against $H_{0},$ then this would indicate a poor diagnostic. If evidence is found in favor, then we can proceed conditionally given that $H_{0}$ holds and so condition the joint prior and joint posterior on this event being true when making inferences about AUC, $c_{opt},$ etc. So for the prior it is necessary to generate $1/\sigma^{2}∼$ gamma$\left(\alpha_{0},\beta_{0}\right)$ and then generate $\left(\mu_{D},\mu_{ND}\right)$ from the joint conditional prior given $\sigma^{2}$ and that $\mu_{D}>\mu_{ND}.$ Denoting the conditional priors given $\sigma^{2}$ by $\pi_{D}\left(\mu_{D}\;|\;\sigma^{2}\right)$ and $\pi_{ND}\left(\mu_{ND}\;|\;\sigma^{2}\right),$ we see that this joint conditional prior is proportional to*

$$\pi_{ND}\left(\mu_{ND}\;|\;\sigma^{2}\right)\pi_{D}\left(\mu_{D}\;|\;\sigma^{2}\right)=\Pi_{ND}\left(\mu_{ND}<\mu_{D}\;|\;\mu_{D},\sigma^{2}\right)\frac{\pi_{ND}\left(\mu_{ND}\right)}{\Pi_{ND}\left(\mu_{ND}<\mu_{D}\;|\;\sigma^{2}\right)}\pi_{D}\left(\mu_{D}\;|\;\sigma^{2}\right).$$

*While generally it is not possible to generate efficiently from this distribution we can use importance sampling to calculate any expectations by generating $\mu_{D}∼\mu_{D}\;|\;\sigma^{2}∼N\left(\mu_{0},\tau_{0}^{2}\sigma^{2}\right),\mu_{ND}∼N\left(\mu_{0},\tau_{0}^{2}\sigma^{2}\;|\;\left(-\infty ,\mu_{D}\right]\right)$ with $\Pi_{ND}\left(\mu_{ND}<\mu_{D}\;|\;\mu_{D},\sigma^{2}\right)=\Phi \left(\left(\mu_{D}-\mu_{0}\right)/\tau_{0}\sigma \right)$ serving as the importance sampling weight and where $N(\mu_{0},\tau_{0}^{2}\sigma^{2}\;|\;\left(-\infty ,\mu_{D}\right])$ denotes the $N(\mu_{0},\tau_{0}^{2}\sigma^{2})$ distribution conditioned to $\left(-\infty ,\mu_{D}\right]$ with density*

$$\Phi^{-1}\left(\left(\mu_{D}-\mu_{0}\right)/\tau_{0}\sigma \right){\left(2\pi \tau_{0}^{2}\sigma^{2}\right)}^{-1/2}\varphi \left(\left(\mu_{ND}-\mu_{0}\right)/\tau_{0}\sigma \right)$$

*for $\mu_{ND}\leq \mu_{D}$ and 0 otherwise. Generating from this distribution via inversion is easy since the cdf is $\Phi \left(\left(\mu_{ND}-\mu_{0}\right)/\tau_{0}\sigma \right)/\Phi \left(\left(\mu_{D}-\mu_{0}\right)/\tau_{0}\sigma \right).$ Note that, if we take the posterior from the unconditioned prior and condition that, we will get the same conditioned posterior as when we use the conditioned prior to obtain the posterior. This implies that in the joint posterior for $\left(\mu_{ND},\mu_{D},\sigma^{2}\right)$ it is only necessary to adjust the posterior for $\mu_{ND}$ as was done with the prior and this is also easy to generate from. Note that Lemma 2 (i) implies that it is necessary to use the conditional prior and posterior to guarantee that $c_{opt}$ exists finitely.*

*Since $H_{0}$ was accepted, the conditional sampling was implemented and the estimate of the AUC is 0.795 with plausible region [0.670,0.880] which has posterior content 0.856. So the estimate is close to the true value but there is substantial uncertainty. Figure 4 is a plot of the conditioned prior, the conditioned posterior and relative belief ratio for this data.*

*With the specified prior for w, the posterior is beta (35.3589,47.53835) which leads to estimate 0.444 for w with plausible interval (0.374,0.516) having posterior probability content 0.782. Using this prior and posterior for w and the conditioned prior and posterior for $(\mu_{D},\mu_{ND},\sigma^{2}),$ we proceed to an inference about $c_{opt}$ and the error characteristics associated with this classification. A computational problem arises when obtaining the prior and posterior distributions of $c_{opt}$ as it is clear from (Equation 5) that these distributions can be extremely long-tailed. As such, we transform to $c_{\mathrm{mod}}=0.5+\arctan\left(c_{opt}\right)/\pi \in \left[0,1\right]$ (the Cauchy cdf), obtain the estimate $c_{\mathrm{mod}}\left(d\right)$ where $d=(n_{ND},{\bar{x}}_{ND},s_{ND}^{2},n_{D},{\bar{x}}_{D},s_{D}^{2})$ and its plausible region and then, applying the inverse transform, obtain $c_{opt}\left(d\right)=\tan\left(\pi \left(c_{\mathrm{mod}}\left(d\right)-0.5\right)\right)$ and its plausible region. It is notable that relative belief inferences are invariant under 1-1 smooth transformations, so it does not matter which parameterization is used, but it is much easier computationally to work with a bounded quantity. Furthermore, if a shorter tailed cdf is used rather than a Cauchy, e.g., a N(0,1) cdf, then errors can arise due to extreme negative values being always transformed to 0 and very extreme positive values always transformed to 1. Figure 5 is a plot of the prior density, posterior density and relative belief ratio of $c_{\mathrm{mod}}.$ For these data $c_{opt}\left(d\right)=0.715$ with plausible interval (0.316,1.228) having posterior content 0.860. Large Monte Carlo samples were used to get smooth estimates of the densities and relative belief ratio but these only required a few minutes of computer time on a desktop. The estimated error characteristics at this value of $c_{opt}$ are as follows: FNR(0.715)=0.41, FPR(0.715)=0.22, Error(0.715)=0.27, FDR(0.715)=0.30, FNDR(0.715)=0.24 which are close to the true values.*


**Example 4.** 
*Binormal with $\sigma_{ND}^{2}\neq \sigma_{D}^{2}.$*

*In this case the prior is given by $\pi_{1}\left(\mu_{ND},\sigma_{ND}^{2}\right)\pi_{2}\left(\mu_{D},\sigma_{D}^{2}\right)$ where*

(10)
$$\begin{array}{cc} \; & \mu_{ND}\;|\;\sigma_{ND}^{2}∼N\left(\mu_{0},\tau_{0}^{2}\sigma_{ND}^{2}\right),\;1/\sigma_{ND}^{2}∼\;\mathrm{gamma}\left(\lambda_{1},\lambda_{2}\right)\\ \; & \mu_{D}\;|\;\sigma_{D}^{2}∼N\left(\mu_{0},\tau_{0}^{2}\sigma_{D}^{2}\right),\;1/\sigma_{D}^{2}∼\;\mathrm{gamma}\left(\lambda_{1},\lambda_{2}\right). \end{array}$$

*Although this specifies the same prior for the two populations, this is easily modified to use different priors and, in any case, the posteriors are different. Again it is necessary to check that the AUC >1/2 but also to check that $c_{opt}$ exists using the full posterior based on this prior and for this we have the hypothesis $H_{0}$ given by Corollary 1. If evidence in favor of $H_{0}$ is found, the prior is replaced by the conditional prior given this event for inference about $c_{opt}.$ This can be implemented via importance sampling as was done in Example 3 and similarly for the posterior.*

*Using the same data and hyperparameters as in Example 3 the relative belief ratio of $H_{0}$ is 3.748 with posterior content 0.828 so there is reasonably strong evidence in favor of $H_{0}.$ Estimating the value of the AUC is then based on conditioning on $H_{0}$ being true. Using the conditional prior given that $H_{0}$ is true, the relative belief estimate of the AUC is 0.793 with plausible interval (0.683,0.857) with posterior content 0.839. The optimal cutoff is estimated as $c_{opt}\left(d\right)=0.739$ with plausible interval (0.316,1.228) having posterior content 0.875. Figure 6 is a plot of the prior density, posterior density and relative belief ratio of $c_{\mathrm{mod}}.$ The estimates of the error characteristics at $c_{opt}\left(d\right)$ are as follows: FNR(0.739)=0.43, FPR(0.739)=0.19, Error(0.739)=0.28, FDR(0.739)=0.28, FNDR(0.624)=0.264.*

*It is notable that these inferences are very similar to those in Example 3. It is also noted that the sample sizes are not big and so the only situation where it might be expected that the inferences will be quite different between the two analyses is when the variances are substantially different.*


### 3.4. Nonparametric Bayes Model

Suppose that *X* is a continuous variable, of course still measured to some finite accuracy, and available information is such that no particular finite dimensional family of distributions is considered feasible. The situation is considered where a normal distribution $N(\mu ,\sigma^{2}),$ perhaps after transforming the data, is considered as a possible base distribution for *X* but we want to allow for deviation from this form. Alternative choices can also be made for the base distribution. The statistical model is then to assume that the $x_{ND}$ and $x_{D}$ are generated as samples from $F_{ND}$ and $F_{D},$ where these are independent values from a DP(a,H) (Dirichlet) process with base $H=N(\mu ,\sigma^{2})$ for some $(\mu ,\sigma^{2})\;$ and concentration parameter a. Actually, since it is difficult to argue for some particular choice of $(\mu ,\sigma^{2}),$ it is supposed that $\left(\mu ,\sigma^{2}\right)$ also has a prior $\pi (\mu ,\sigma^{2}).$ The prior on $\left(F_{ND},F_{D}\right)$ is then specified hierarchically as a mixture Dirichlet process,
$$\begin{array}{cc} \; & \left(\mu_{ND},\sigma_{ND}^{2}\right)∼\pi \;\mathrm{independent}\;\mathrm{of}\;\;\left(\mu_{D},\sigma_{D}^{2}\right)∼\pi ,\\ \; & F_{ND}\;|\;\left(\mu_{ND},\sigma_{ND}^{2}\right)∼\;\mathrm{DP}\left(a_{ND},N\left(\mu_{ND},\sigma_{ND}^{2}\right)\right)\;\mathrm{independent}\;\mathrm{of}\;\\ \; & F_{D}\;|\;\left(\mu_{D},\sigma_{D}^{2}\right)∼\;\mathrm{DP}\left(a_{D},N\left(\mu_{D},\sigma_{D}^{2}\right)\right). \end{array}$$To complete the prior it is necessary to specify π and the concentration parameters $a_{ND}$ and $a_{D}.$ For π the prior is taken to be a normal distribution elicited as discussed in Section 3.3 although other choices are possible. For eliciting the concentration parameters, consider how strongly it is believed that normality holds and for convenience suppose $a=a_{ND}=a_{D}.$ If F∼ DP(a,H) with *H* a probability measure, then E(F(A))=H(A) and Var(F(A))=H(A)(1−H(A))/(1+a). When *F* a random measure from P, then $\sup_{A}P\left(|F\left(A\right)-H\left(A\right)|\geq \varepsilon \right)=\sup_{A}\left\{1-P\left(\max\left(0,H\left(A\right)-\varepsilon \right)<F\left(A\right)<\min\left(1,H\left(A\right)+\varepsilon \right)\right)\right\}$ which, when P∼ DP(a,H), equals
(11)$$\underset{r\in [0,1]}{\sup}\left\{1-B\left(\left[\max\left(0,r-\varepsilon \right),\min\left(1,r+\varepsilon \right)\right],ar,a\left(1-r\right)\right)\right\}$$
where $B(\cdot ,\beta_{1},\beta_{2})$ denotes the beta$\left(\beta_{1},\beta_{2}\right)$ measure. This upper bound on the probability that the random *F* differs from *H* by at least ε on an event can be made as small as desirable by choosing *a* large enough. For example, if ε=0.25 and it is required that this upper bound be less than 0.1, then this satisfied when a≥9.8 and if instead ε=0.1, then a≥66.8 is necessary. Note that, since this bound holds for every continuous probability measure H, it also holds when *H* is random, as considered here. So *a* is controlling how close it is believed that the true distribution is to *H*. Alternative methods for eliciting *a* can be found in [24,25].

Generating $\left(F_{ND},F_{D}\right)$ from the prior for given (a,H) can only be done approximately and the approach of [26] is adopted. For this, integer $n^{*}$ is specified and measure $P_{n^{*}}=\sum_{i=1}^{n^{*}}p_{i,n^{*}}I_{\left\{c_{i}\right\}}$ is generated where $\left(p_{1,n^{*}},\ldots ,p_{n^{*},n^{*}}\right)∼Dirichlet\left(a/n^{*},\ldots ,.a/n^{*}\right)$ independent of $c_{1},\ldots ,c_{n^{*}}\overset{iid}{∼}H,$ since $P_{n^{*}}\overset{w}{\to }$ DP(a,H) as $n^{*}\to \infty .$ So to carry out a priori calculations proceed as follows. Generate
$$\begin{array}{cc} \; & \left(p_{ND1,n^{*}},\ldots ,p_{NDn^{*},n^{*}}\right)∼\mathrm{Dirichlet}\left(\left(a/n^{*}\right)1_{n^{*}}\right),\;\left(\mu_{ND},\sigma_{ND}^{2}\right)∼\pi ,\\ \; & \left(c_{ND1},\ldots ,c_{NDn^{*}}\right)\;|\;\left(\mu_{ND},\sigma_{ND}^{2}\right)\overset{i.i.d.}{∼}N\left(\mu_{ND},\sigma_{ND}^{2}\right),\;w∼\mathrm{beta}\left(\alpha_{1w},\alpha_{2w}\right) \end{array}$$
and similarly for $\left(p_{D1,n^{*}},\ldots ,p_{Dn^{*},n^{*}}\right),\left(\mu_{D},\sigma_{D}^{2}\right),$ and $(c_{D1},\ldots ,c_{Dn^{*}}).$ Then $F_{ND,n^{*}}\left(c\right)=\sum_{\left\{i:c_{NDi}\leq c\right\}}p_{NDin^{*}}$ is the random cdf at $c\in R^{1}$ and similarly for $F_{D,n^{*}}$, so AUC $=\sum_{i=1}^{n^{*}}\left(1-F_{D,n^{*}}\left(c_{NDi}\right)\right)p_{NDi,n^{*}}\;$ is a value from the prior distribution of the AUC. This is done repeatedly to get the prior distribution of the AUC as in our previous discussions and we proceed similarly for the other quantities of interest.

Now $F_{ND}\;|\;x_{ND},\left(\mu_{ND},\sigma_{ND}^{2},\mu_{D},\sigma_{D}^{2}\right)∼\mathrm{DP}\left(a+n_{ND},H_{ND}\right)$ independent of $F_{D}\;|\;x_{D},\left(\mu_{ND},\sigma_{ND}^{2},\mu_{D},\sigma_{D}^{2}\right)∼\mathrm{DP}\left(a+n_{D},H_{D}\right)$ with $H_{ND}\left(c\right)=a\Phi \left(\left(c-\mu_{ND}\right)/\sigma_{ND}\right)/\left(a+n_{ND}\right)+n_{ND}{\hat{F}}_{ND}\left(c\right)/\left(a+n_{ND}\right)$ and ${\hat{F}}_{ND}\left(c\right)=\sum_{i=1}^{n_{ND}}I_{\left(-\infty ,c\right]}\left(x_{NDi}\right)/n_{ND}$ is the empirical cdf (ecdf) based on $x_{ND}$ and similarly for $H_{D}.$ The posteriors of $\left(\mu_{ND},\sigma_{ND}^{2}\right)$ and $\left(\mu_{D},\sigma_{D}^{2}\right)$ are obtained via results in [27,28]. The posterior density of $\left(\mu_{ND},\sigma_{ND}^{2}\right)$ given $x_{ND}$ is proportional to
$$\pi \left(\mu_{ND},\sigma_{ND}^{2}\right)\prod_{i=1}^{{\tilde{n}}_{ND}}\sigma_{ND}^{-1}\varphi \left(\left({\tilde{x}}_{NDi}-\mu_{ND}\right)/\mu_{ND}\right)$$
where ${\tilde{n}}_{ND}$ is the number of unique values in $x_{ND}$ and $\left\{{\tilde{x}}_{ND1},\ldots ,{\tilde{x}}_{ND{\tilde{n}}_{ND}}\right\}$ is the set of unique values with mean ${\tilde{x}}_{ND}$ and sum of squared deviations ${\tilde{s}}_{ND}^{2}$. From this it is immediate that
$$\begin{array}{cc} \; & \mu_{ND}\;|\;\sigma_{ND}^{2},x_{ND}∼N\left({\left({\tilde{n}}_{ND}+1/\tau_{0}^{2}\right)}^{-1}\left({\tilde{n}}_{ND}{\tilde{x}}_{ND}+\mu_{0}/\tau_{0}^{2}\right),{\left({\tilde{n}}_{NDND}+1/\tau_{0}^{2}\right)}^{-1}\sigma_{ND}^{2}\right),\\ \; & 1/\sigma_{ND}^{2}\;|\;x_{ND}∼\;\mathrm{gamma}\left(\alpha_{0}+{\tilde{n}}_{ND}/2,{\tilde{\lambda }}_{x_{ND}}\right) \end{array}$$
where ${\tilde{\lambda }}_{x_{ND}}=\lambda_{0}+{\tilde{s}}_{ND}^{2}/2+{\left({\tilde{n}}_{ND}+1/\tau_{0}^{2}\right)}^{-1}\left({\tilde{n}}_{ND}/\tau_{0}^{2}\right){\left({\tilde{x}}_{ND}-\mu_{0}\right)}^{2}/2.$ A similar result holds for the posterior of $(\mu_{D},\sigma_{D}^{2}).$

To approximately generate from the full posterior specify some $n^{**},$ put $p_{a,n_{ND}}=a/\left(a+n_{ND}\right),q_{a,n_{ND}}=1-p_{a,n_{ND}}$ and generate
$$\begin{array}{cc} \; & \left(p_{ND1,n^{**}},\ldots ,p_{NDn^{**},n^{**}}\right)\;|\;x_{ND}∼\;\mathrm{Dirichlet}\left(\left(\left(a+n_{ND}\right)/n^{**}\right)1_{n^{**}}\right),\\ \; & \left(\mu_{ND},\sigma_{ND}^{2}\right)\;|\;x_{ND}∼\pi \left(\cdot \;|\;x_{ND}\right),\\ \; & \left(c_{ND1},\ldots ,c_{NDn^{**}}\right)\;|\;\left(\mu_{ND},\sigma_{ND}^{2}\right),x_{ND}\overset{i.i.d.}{∼}p_{a,n_{ND}}N\left(\mu_{ND},\sigma_{ND}^{2}\right)+q_{a,n_{ND}}{\hat{F}}_{ND},\\ \; & w\;|\;x_{ND}∼\;\mathrm{beta}\left(\alpha_{1w}+n_{D},\alpha_{2w}+n_{ND}\right) \end{array}$$
and similarly for $\left(p_{D1,n^{**}},\ldots ,p_{Dn^{**},n^{**}}\right),\left(\mu_{D},\sigma_{D}^{2}\right)$ and $(c_{D1},\ldots ,c_{Dn^{**}}).$ If the data does not comprise a sample from the full population, then the posterior for *w* is replaced by its prior.

There is an issue that arises when making inference about $c_{opt},$ namely, the distributions for $c_{opt}$ that arises from this approach can be very irregular and particularly the posterior distribution. In part this is due to the discreteness of the posterior distributions of $F_{ND}$ and $F_{D}$. This does not affect the prior distribution because the points on which the generated distributions are concentrated vary quite continuously among the realizations and this leads to a relatively smooth prior density for $c_{opt}.$ For the posterior, however, the sampling from the ecdf leads to a very irregular, multimodal density for $c_{opt}.$ So some smoothing is necessary in this case.

Consider now applying such an analysis to the dataset of Example 3, where we know the true values of the quantities of interest and then to a dataset concerned with the COVID-19 epidemic.

**Example 5.** 
*Binormal data (Examples 3 and 4)*

*The data used in Example 3 are now analyzed but using the methods of this section. The prior on $\left(\mu_{ND},\sigma_{ND}^{2}\right),\left(\mu_{D},\sigma_{D}^{2}\right)$ and w is taken to be the same as that used in Example 4 so the variances are not assumed to be the same. The value ε=0.25 is used and requiring (Equation 11) to be less than 0.018 leads to a=20. So the true distributions are allowed to differ quite substantially from a normal distribution. Testing the hypothesis $H_{0}:$ AUC >1/2 led to the relative belief ratio 1.992 (maximum possible value is 2) and the strength of the evidence is 0.997 so there is strong evidence that $H_{0}$ is true. The AUC, based on the prior conditioned on $H_{0}$ being true, is estimated to be equal to 0.839 with plausible interval (0.691,0.929) having posterior content 0.814. For these data $c_{opt}\left(d\right)=0.850$ with plausible interval (0.45,1.75) having posterior content 0.835. The true value of the AUC is 0.760 and the true value of $c_{opt}$ is 0.905 so these inferences are certainly reasonable although, as one might expect, when the length of the plausible intervals are taken into account, they are not as accurate as those when binormality is assumed as this is correct for this data. So the DP approach worked here although the posterior density for $c_{opt}$ was quite multimodal and required some smoothing (averaging 3 consecutive values).*


**Example 6.** 
*COVID-19 data.*

*A dataset was downloaded from https://github.com/YasinKhc/Covid-19 containing data on 3397 individuals diagnosed with COVID-19 and includes whether or not the patient survived the disease, their gender and their age. There are 1136 complete cases on these variables of which 646 are male, with 52 having died, and 490 are female, with 25 having died. Our interest is in the use of a patient’s age X to predict whether or not they will survive. More detail on this dataset can be found in [29]. The goal is to determine a cutoff age so that extra medical attention can be paid to patients beyond that age. Furthermore, it is desirable to see whether or not gender leads to differences so separate analyses can be carried out by gender. So, for example, in the male group ND refers to those males with COVID-19 that will not die and D refers to the population that will. Looking at histograms of the data, it is quite clear that binormality is not a suitable assumption and no transformation of the age variable seems to be available to make a normality assumption more suitable. Table 3 gives summary statistics for the subgroups. Of some note is that condition (Equation 8), when using standard estimates for population quantities such as w=52/646=0.08 for Males and w=25/490=0.05 for females, is not satisfied which suggests that in a binormal analysis no finite optimal cutoff exists.*

*For the prior, it is assumed that $\left(\mu_{ND},\sigma_{ND}^{2}\right)$ and $\left(\mu_{D},\sigma_{D}^{2}\right)$ are independent values from the same prior distribution as in (Equation 10). For the prior elicitation, as discussed in Example 3, suppose it is known with virtual certainty that both means lie in (20,70) and $\left(l_{0},u_{0}\right)=\left(20,50\right)$ so we take $\mu_{0}=45,\tau_{0}=\left(m_{2}-m_{1}\right)/2u_{0}=0.75$ and the iterative process leads to $\left(\lambda_{1},\lambda_{2}\right)=\left(8.545,1080.596\right)$ which implies a prior on the σ’s with mode at 10.932 and the interval (7.764,19.411) containing 0.99 of the prior probability. Here the relevant prevalence refers to the proportion of COVID-19 patients that will die and it is supposed that w∈[0.00,0.15] with virtual certainty which implies w∼ beta(9.81,109.66). So the prior probability that someone with COVID-19 will die is assumed to be less than 15% with virtual certainty. Since normality is not an appropriate assumption for the distribution of X, the choice ε=0.25 with the upper bound (Equation 11) equal to 0.1 seems reasonable and so a=9.8. This specifies the prior that is used for the analysis with both genders and it is to be noted that it is not highly informative.*

*For males the hypothesis AUC >1/2 is assessed and RB=1.991 (maximum value 2) with strength effectively equal to 1.00 was obtained, so there is extremely strong evidence that this is true. The unconditional estimate of the AUC is 0.808 with plausible region [0.698,0.888] having posterior content 0.959, so there is a fair bit of uncertainty concerning the true value. For the conditional analysis, given that AUC >1/2, the estimate of the AUC is 0.806 with plausible region [0.731,0.861] having posterior content 0.932. So the conditional analysis gives a similar estimate for the AUC with a small increase in accuracy. In either case it seems that the AUC is indicating that age should be a reasonable diagnostic. Note that the standard nonparametric estimate of the AUC is 0.810 so the two approaches agree here. For females the hypothesis AUC >1/2 is assessed and RB=1.994 with strength effectively equal to 1 was obtained, so there is extremely strong evidence that this is true. The unconditional estimate of the AUC is 0.873 with plausible region (0.742,0.948) having posterior content 0.968. For the conditional analysis, given that AUC >1/2, the estimate of the AUC is 0.874 with plausible region (0.791,0.936) having posterior content 0.956. The traditional estimate of the AUC is 0.902 so the two approaches are again in close agreement.*

*Inferences for $c_{opt}$ are more problematical in both genders. Consider the male data. The data set is very discrete as there are many repeats and the approach samples from the ecdf about 84% of the time for the males that died and 98% of the time for the males that did not die. The result is a plausible region that is not contiguous even with smoothing. Without smoothing the estimate is $c_{opt}\left(d\right)=85.5$ for males, which is a very dominant peak for the relative belief ratio. The plausible region contains 0.928 of the posterior probability and, although it is not a contiguous interval, the subinterval [85.2,85.8] is a 0.58-credible interval for $c_{opt}$ that is in agreement with the evidence. If we make the data continuous by adding a uniform(0,1) random error to each age in the data set, then $c_{opt}\left(d\right)=86.1$ and plausible interval [75.9,86.7] with posterior content 0.968 is obtained. These cutoffs are both greater than the maximum value in the ND data, so there is ample protection against false positives but it is undoubtedly false negatives that are of most concern in this context. If instead the FNDR is used as the error criterion to minimize, then $c_{opt}\left(d\right)=35.7$ and plausible interval [26.1,35.7] with posterior content 0.826 is obtained and so in this case there will be too many false positives. So a useful optimal cutoff incorporating the relevant prevalence does not seem to exist with these data.*

*If the relevant prevalence is ignored and $w_{0}$FNR$+(1-w_{0})$FPR is used for some fixed weight $w_{0}$ to determine $c_{opt}\left(d\right)$, then more reasonable values are obtained. Table 4 gives the estimates for various $w_{0}$ values. With $w_{0}=0.5$ (corresponding to using Youden’s index) $c_{opt}\left(d\right)=65.7$ while if $w_{0}=0.7,$ then $c_{opt}\left(d\right)=56.7.$ When $w_{0}$ is too small or too large then the value of $c_{opt}\left(d\right)$ is not useful. While these estimates do not depend on the relevant prevalence, the error characteristics that do depend on this prevalence (as expressed via its prior and posterior distributions) can still be quoted and a decision made as to whether or not to use the diagnostic. Table 5 contains the estimates of the error characteristics at $c_{opt}\left(d\right)$ for various values of $w_{0}$ where these are determined using the prior and posterior on the relevant prevalence w. Note that these estimates are determined as the values that maximize the corresponding relative belief ratios and take into account the posterior of w. So, for example, the estimate of the Error is not the convex combination of the estimates of FNR and FPR based on the $w_{0}$ weight. Another approach is to simply set the cutoff Age at a value at a value $c_{0}$ and then investigate the error characteristics at that value. For example, with $c_{0}=60,$ then the estimated values are given by FNR$(c_{0})=0.238,$ FPR$(c_{0})=0.308,$ Error$(c_{0})=0.328,$ FDR$(c_{0})=$ 0.818 and FNDR$(c_{0})=0.028.$*

*Similar results are obtained for the cutoff with female data although with different values. Overall, Age by itself does not seem to be useful classifier although that is a decision for medical practitioners. Perhaps it is more important to treat those who stand a significant chance of dying more extensively and not worry too much that some treatments are not necessary. The clear message from this data, however, is that a relatively high AUC does not immediately imply that a diagnostic is useful and the relevant prevalence is a key aspect of this determination.*


## 4. Conclusions

ROC analyses represent a significant practical application of statistical methodology. While previous work has considered such analyses within a Bayesian framework, this has typically required the specification of loss functions. Losses are not required in the approach taken here which is entirely based on a natural characterization of statistical evidence via the principle of evidence and the relative belief ratio. As discussed in Section 2.2 this results in a number of good properties for the inferences that are not possessed by inferences derived by other approaches. While the Bayes factor is also a valid measure of evidence, its usage is far more restricted than the relative belief ratio which can be applied with any prior, without the need for any modifications, for both hypothesis assessment and estimation problems. This paper has demonstrated the application of relative belief to ROC analyses under a number of model assumptions. In addition, as documented in points (ii)–(vi) of the Introduction, a number of new results have been developed for ROC analyses more generally.

## Figures and Tables

**Figure 1 entropy-24-01710-f001:**
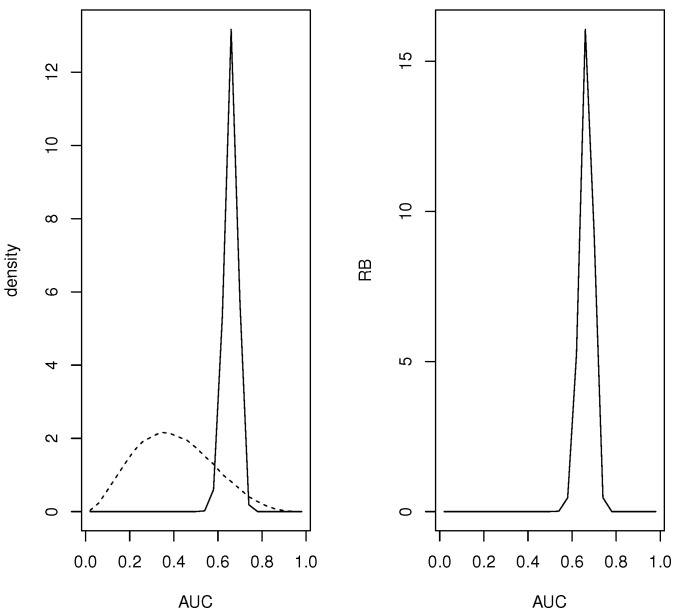
In Example 2, plots of the prior (- - -), the posterior (—) and the RB ratio of the AUC.

**Figure 2 entropy-24-01710-f002:**
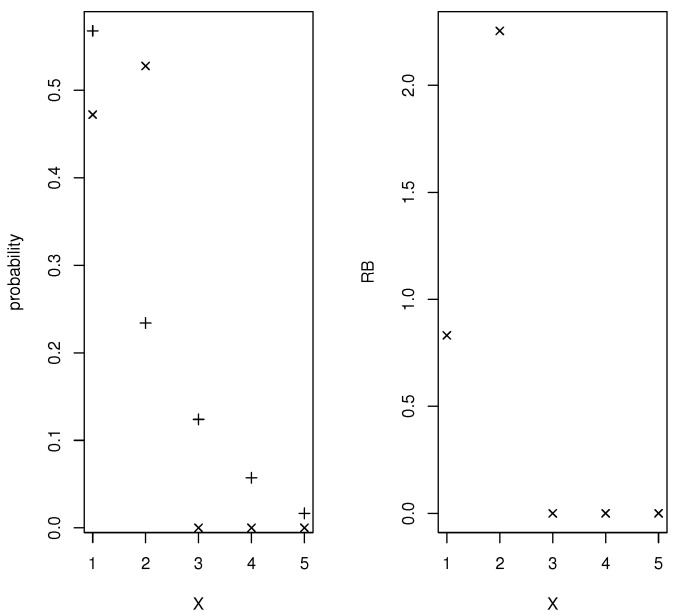
In Example 2, plots of the the prior (+), the posterior (×) and the RB ratio of $c_{opt}$.

**Figure 3 entropy-24-01710-f003:**
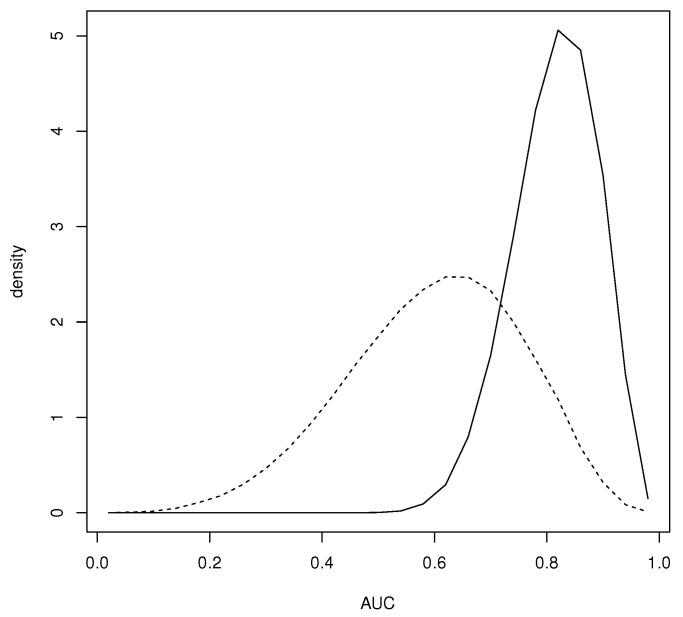
Prior density of the AUC when $p_{D}$ is uniform on the set of nondecreasing probabilities independent of $p_{ND}$ uniform on the set of nonincreasing probabilities (–) as well as when $p_{D}$ is uniformly distributed on the set of nondecreasing probabilities independent of $p_{ND}$ uniform on $S_{k}$ (- -).

**Figure 4 entropy-24-01710-f004:**
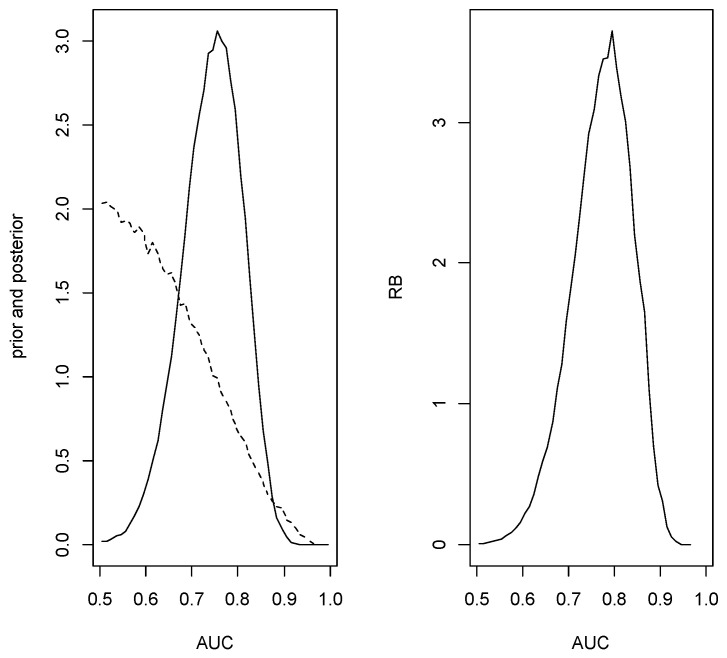
The conditioned prior (- -) and posterior (–) densities (left panel) and the relative belief ratio (right panel) of the AUC in Example 3.

**Figure 5 entropy-24-01710-f005:**
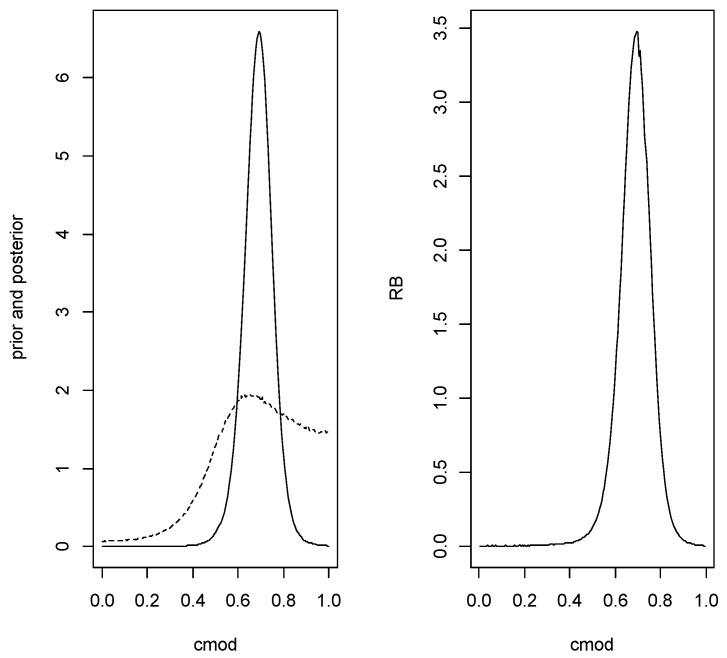
Plots of the prior (- -), posterior (left panel) and relative belief ratio (right panel) of $c_{opt}$ in Example 3.

**Figure 6 entropy-24-01710-f006:**
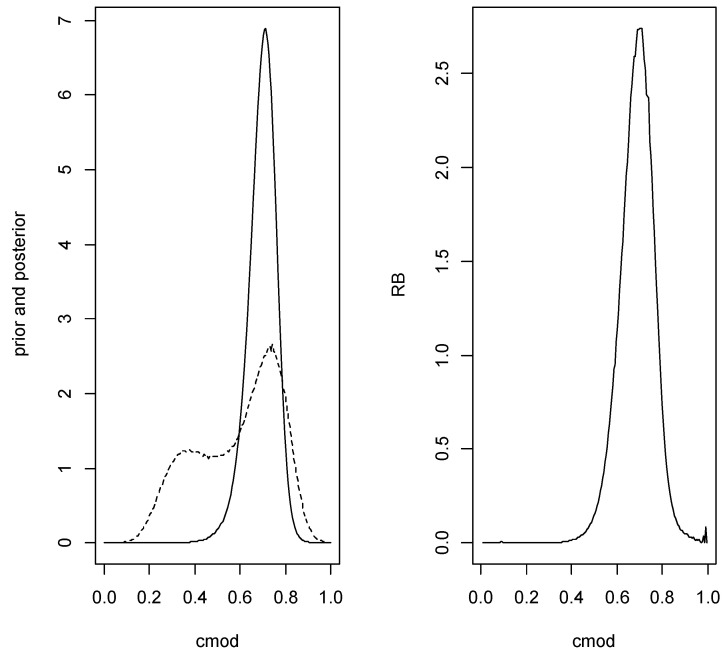
Plots of the prior (- -), posterior (left panel) and relative belief ratio (right panel) of $c_{opt}$ in Example 4.

**Table 1 entropy-24-01710-t001:** Error probabilities when X>c indicates a positive.

	$\Omega_{D}$	$\Omega_{ND}$
X>c	$\begin{array}{c} \mathrm{TPR}\left(c\right)=1-F_{D}\left(c\right)\\ \mathrm{sensitivity}\;(\mathrm{recall})\;\mathrm{or}\\ \mathrm{true}\;\mathrm{positive}\;\mathrm{rate} \end{array}$	$\begin{array}{c} \mathrm{FPR}\left(c\right)=1-F_{ND}\left(c\right)\\ \mathrm{false}\;\mathrm{positive}\;\mathrm{rate} \end{array}$
X≤c	$\begin{array}{c} \mathrm{FNR}\left(c\right)=F_{D}\left(c\right)\\ \mathrm{false}\;\mathrm{negative}\;\mathrm{rate} \end{array}$	$\begin{array}{c} \mathrm{TNR}\left(c\right)=F_{ND}\left(c\right)\\ \mathrm{specificity}\;\mathrm{or}\\ \mathrm{true}\;\mathrm{negative}\;\mathrm{rate} \end{array}$

**Table 2 entropy-24-01710-t002:** The estimates of the error characteristcs of *X* at $c_{opt}=2$ in Example 2 where (a) *w* is assumed known, (b) only the prior for *w* is available, (c) the posterior for *w* is also available.

Quantity	Estimate (a)	Estimate (b)	Estimate (c)
$\mathrm{FPR}(c_{opt})$	0.30	0.26	0.30
$\mathrm{FNR}(c_{opt})$	0.22	0.22	0.22
$\mathrm{Error}(c_{opt})$	0.22	0.22	0.22
$\mathrm{FDR}(c_{opt})$	0.14	0.14	0.14
$\mathrm{FNDR}(c_{opt})$	0.34	0.34	0.34

**Table 3 entropy-24-01710-t003:** Summary statistics for the data in Example 6.

Group	Number	Mean	std. dev.	Min	Max
ND males	594	48.81	17.72	0.50	85.00
D males	52	68.46	13.66	36.00	89.00
ND females	465	48.69	18.73	2.00	96.00
D females	25	77.36	12.12	48.00	95.00

**Table 4 entropy-24-01710-t004:** Weighted error $w_{0}$FNR+($1-w_{0})$FPR determining $c_{opt}\left(d\right)$ for Males in Example 6.

$w_{0}=$ Weight of FNR	$c_{\mathrm{opt}}\left(d\right)$	Plausible Range (post. prob.)
0.1	85.5	75.3–118.5 (0.945)
0.3	65.1	64.5–85.5 (0.868)
0.5	65.1	55.5–72.3 (0.939)
0.7	56.7	35.7–58.5 (0.919)
0.9	35.7	33.3–52.5 (0.875)

**Table 5 entropy-24-01710-t005:** Error characteristics for Males in Example 6 at various weights.

$w_{0}=$ Weight of FNR	FNR	FPR	Error	FDR	FNDR
0.1	0.918	0.008	0.008	0.458	0.073
0.3	0.368	0.183	0.213	0.733	0.043
0.5	0.368	0.183	0.213	0.733	0.038
0.7	0.158	0.358	0.363	0.823	0.018
0.9	0.003	0.753	0.688	0.893	0.003

## Data Availability

The data and R code used for the examples in Section 3.2, Section 3.3 and Section 3.4 can be obtained at https://utstat.utoronto.ca/mikevans/software/ROCcodeforexamples.zip (accessed on 15 November 2022).

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
