# Peer review of "ROC Analyses Based on Measuring Evidence Using the Relative Belief Ratio"

_entropy, 2022, doi:10.3390/e24121710_

Round 1

Reviewer 1 Report

Nice paper. The authors work on certain ROC analyses based on measuring evidence using relative belief. Nonlinearity should be the bright spot. Their results seem new and they have an impressive list of relevant publications already. I think I should recommend the paper be accepted, only after the following modifications:

A. Physics/Nonlinearity behind the issue and/or around their results should be addressed clearer or in more details, e.g., in the Keywords, Abstract, Introduction and Conclusion.

B. Right before Section 2, write clearly that others' published papers do not cover theirs, i.e., make clear that the paper is really new.

C. Improve the English writing,

e.g., the Introduction, every sentence should be followed by references, especially the newly-published ones.

e.g., "ROC Analyses Based on Measuring Evidence Using Relative Belief" could be changed to "ROC Analyses Based on the Measuring Evidence Using the Relative Belief"  ...  write out the whole words of ROC, AUC, etc.

e.g., other bad English writing ... samples ...

Inferences are derived for the AUC as well as the cutoff c used for classification and the associated error characteristics -> Inferences are derived for the AUC as well as the cutoff c used for classification and the associated error characteristics.

While there are many methods available for the choice of ... -> While there are some methods available for the choice of ...

They are asked to, correspondingly, check the whole paper.

D. The authors need to discuss several relevant new nonlinear-physics papers which have made some meaningful contributions: Commun. Theor. Phys. 72, 095002 (2020); China Ocean Eng. 35, 518-530 (2021); Appl. Math. Lett. 120, 107161 (2021); Chaos Solitons Fract. 162, 112486 (2022); Qual. Theory Dyn. Syst. 21, 95 (2022); Chaos Solitons Fract. 164, 112672 (2022). Also, please address other relevant new nonlinear-physics work in their paper: Nonlinear Dyn. 108, 2417-2428 (2022); Appl. Math. Lett. 122, 107301 (2021); Appl. Math. Lett. 128, 107858 (2022).

(Below .. not required but suggested ...)

E.  The results could be, mathematically speaking, written as the theorems.

F.  Are there any comparisons with physics experiments and/or observations?  Say something here.

G. Say something more on the numerical.

Reviewer 2 Report

In this paper, the authors study ROC analysis, in particular inference of the AUC, cutoff, and error characteristics. They introduce relative belief ratio for the estimation of the quantities of interests.

The study contains important perspective and discussions, but the writing needs to be improved. 

In the introduction, they are explaining related previous studies, but it is difficult to understand what are the unsolved problems in these studies, and how this study approaches to these problems. I would recommend that you devise a way to write this paper so that readers can distinguish between the story of the previous study and that of this paper. 

Technically, the relative belief ratio, which the authors proposed, is similar to the Bayes factor or the odds ratio, in the sense that both of them consider the ratio between the posterior and the prior. An explanation of the differences between these existing and proposed methods is needed.

In the sections 2. and 3., analytical methods are explained, but the novelty of the method is difficult to understand. Comparisons between the methods proposed in previous research are required, to show the validness of the proposed methods.

It is also difficult to understand what is claiming in the conclusion.

Round 2

Reviewer 2 Report

The revised manuscript correctly explains the details of their research and related research fields.

This paper is worthy of publication.

Author Response

Thank-you